Resource

# Concordance of MERFISH spatial transcriptomics with bulk and single-cell RNA sequencing

Jonathan Liu[1],* , Vanessa Tran[1],*, Venkata Naga Pranathi Vemuri[1], Ashley Byrne[1] , Michael Borja[1], Yang Joon Kim[1] , Snigdha Agarwal[1], Ruofan Wang[1], Kyle Awayan[1], Abhishek Murti[2] , Aris Taychameekiatchai[2], Bruce Wang[2] , George Emanuel[3], Jiang He[3], John Haliburton[1], Angela Oliveira Pisco[1] , Norma F Neff[1] 

Spatial transcriptomics extends single-cell RNA sequencing (scRNA-seq) by providing spatial context for cell type identification and analysis. Imaging-based spatial technologies such as multiplexed error-robust fluorescence in situ hybridization (MERFISH) can achieve single-cell resolution, directly mapping single-cell identities to spatial positions. MERFISH produces a different data type than scRNA-seq, and a technical comparison between the two modalities is necessary to ascertain how to best integrate them. We performed MERFISH on the mouse liver and kidney and compared the resulting bulk and single-cell RNA statistics with those from the Tabula Muris Senis cell atlas and from two Visium datasets. MERFISH quantitatively reproduced the bulk RNA-seq and scRNA-seq results with improvements in overall dropout rates and sensitivity. Finally, we found that MERFISH independently resolved distinct cell types and spatial structure in both the liver and kidney. Computational integration with the Tabula Muris Senis atlas did not enhance these results. We conclude that MERFISH provides a quantitatively comparable method for single-cell gene expression and can identify cell types without the need for computational integration with scRNA-seq atlases.

## Introduction

Named "method of the year" by *Nature Methods* in 2021 (Marx, 2021), spatial transcriptomics promises to revolutionize biological investigation by allowing researchers to study cells' transcriptomes in their native spatial context. Recently, a plethora of various spatial transcriptomics techniques have arisen, including platforms such as Slide-Seq (Stickels et al, 2021), 10× Visium (Ståhl et al, 2016), Seq-FISH (Eng et al, 2019), MERFISH (Chen et al, 2015), STARmap (Wang et al, 2018), and GeoMX Digital Spatial Profiler (Merritt et al, 2020), to name a few examples. These techniques all involve different fundamental mechanisms, ranging from traditional RNA sequencing performed on spatially barcoded chips to single-molecule fluorescence microscopy (Moses & Pachter, 2022).

A microscopy-based method, MERFISH provides a particularly compelling approach to complement preexisting single-cell RNA sequencing (scRNA-seq) techniques. Although it can only study hundreds to a few thousands of genes, MERFISH makes up for this limitation by offering single-molecule capability. This results in subcellular spatial resolutions that are orders of magnitude finer than sequencing-based spatial approaches, which are currently limited to spatial resolutions at the length scale of a single-cell or greater. Thus, MERFISH can deeply study a target list of genes of interest with spatial context, drawing upon and complementing preexisting insights offered by traditional single-cell analysis.

Because MERFISH is still relatively new, it lacks a systematic technical comparison with RNA sequencing technologies, especially at the single-cell level. In addition, although MERFISH has provided numerous insights in cell culture (Chen et al, 2015; Moffitt et al, 2016a; Moffitt et al, 2016b; Xia et al, 2019), only recently has MERFISH been successfully used in tissue samples, notably in the mouse brain (Moffitt et al, 2018; Wang et al, 2018; Wang et al, 2020 *Preprint*; Zhang et al, 2020 *Preprint*) and fetal liver (Liu et al, 2020; Lu et al, 2021). Evaluating its potential for further biomedical research necessitates a technical analysis in other types of tissue. As MERFISH relies on a limited probe panel of marker genes, it is crucial to ascertain if MERFISH measurements are independently sufficient and informative to conduct single-cell bioinformatics analysis such as cell type clustering and identification or if they need to be augmented with reference scRNA-seq atlases using computational methods.

In this study, we used the Vizgen MERSCOPE Platform to conduct a technical comparison between MERFISH and RNA sequencing in mouse liver and kidney tissues. By comparing the insights from

[1]Chan Zuckerberg Biohub, San Francisco, CA, USA   [2]School of Medicine, University of California, San Francisco, CA, USA   [3]Vizgen Inc., Cambridge, MA, USA

Correspondence: norma.neff@czbiohub.org
Ashley Byrne's present address is Department of Microchemistry, Proteomics, Lipidomics and Next Generation Sequencing, Genentech, Inc., South San Francisco, CA, USA.
John Haliburton's present address is Prolific Machines, San Francisco, CA, USA.
*Jonathan Liu and Vanessa Tran contributed equally to this work.

MERFISH with traditional bulk and scRNA-seq results from Tabula Muris Senis (Schaum et al, 2020; Tabula Muris Consortium, 2020) and with data from two Visium spatial transcriptomics datasets (Dixon et al, 2022; Guilliams et al, 2022), we explored the advantages and limitations of MERFISH for bioinformatics analysis. After establishing that the statistics exhibited by both imaging and sequencing modalities were fairly similar, with MERFISH exhibiting superior dropout rates and sensitivity, we examined MERFISH's ability to identify different cell types. Intriguingly, MERFISH was able to distinguish between different cell types in liver and kidney tissue with standard single-cell bioinformatics analysis, even with only a 307-gene panel containing marker genes split between three organs. Furthermore, we were able to resolve clear structure in the spatial distribution of these cell types, for example, in spatial patterning of pericentral and periportal hepatocytes and endothelial cells in the mouse liver and of podocytes in the mouse kidney.

We then integrated our MERFISH measurements with Tabula Muris Senis using a few computational methods (Lopez et al, 2018; Korsunsky et al, 2019; Kang et al, 2021; Xu et al, 2021) to automatically predict cell type annotations and investigate what additional insights were offered via computational integration. Interestingly, computational integration did not enhance cell type identification, indicating that MERFISH data alone can sufficiently resolve distinct cell types. Although most cell types were similarly labeled by integration or manual annotation, some cell types like podocytes were consistently misclassified by the integration methods. We ascertained that these errors stemmed from intrinsic differences in RNA statistics between MERFISH and scRNA-seq, rather than from the integration methods themselves.

# Results

MERFISH uses a combination of single-molecule fluorescence in situ hybridization (smFISH) and combinatorial labeling of RNA transcripts with optical barcoding to achieve a highly multiplexed, single-molecule readout of transcriptional activity in fixed samples (Chen et al, 2015). Here, we briefly describe the essence of the technique as applied in this work.

MERFISH provides targeted labeling of RNA transcripts by using a preselected gene panel, where each transcript is assigned a unique binary barcode (Fig S1A). These barcodes are error robust, allowing for reliable transcript identification even with several hundreds of genes (Moffitt et al, 2016b). After hybridizing a sample with encoding probes that effectively imprint the desired barcodes onto each RNA species, the barcode is then detected by sequential rounds of multichannel imaging, flowing in different subsets of fluorescently labeled readout probes to hybridize with the barcode region of encoding probes. Fluorescent spots are computationally decoded into their respective binary barcodes wherein the presence of a spot indicates a "1" and the absence indicates a "0" (Fig S1A). These barcodes and their intracellular positions are combined with cell nucleus and boundary staining, allowing for segmentation and measurement of gene expression at single-cell resolution.

Fig S1B summarizes the experimental workflow. After designing a gene panel barcode scheme, we hybridized encoding probes to a tissue sample and imaged it (see the Materials and Methods section for detailed overview of protocol). The resulting raw images contained the necessary information to decode RNA transcript counts. Staining with a combination of DAPI and membrane protein antibodies produced a fluorescent readout on cell nuclei and boundaries. Using the MERlin image analysis and cellpose segmentation packages (Emanuel et al, 2020; Stringer et al, 2021), these raw images were processed to obtain positions of single-RNA transcripts and segmented geometries of individual cells, respectively. These data were then post-processed to calculate more parsimonious summary statistics, such as individual cell areas and single-cell count matrices of each RNA species.

In this work, we investigated a 307-gene panel in the mouse liver and kidney. The genes were selected to be differentially expressed cell type marker genes for the mouse liver, kidney, or pancreas (see the Materials and Methods section for more details). Fourteen datasets were collected from the mouse liver (5), kidney (4), and pancreas (5). Of these, all five of the pancreas samples and two of the kidney samples did not yield enough RNA for downstream analysis (see the Materials and Methods section and Fig S2). Furthermore, one of the kidney samples only yielded enough transcripts for bulk RNA analysis. In addition, only one of the liver samples was stained for DAPI and cell boundary antibodies that allowed for single-cell segmentation. The other four were used for bulk RNA analysis only. Thus, the bulk RNA analysis in this study used five liver and two kidney samples, and the single-cell RNA analysis used one liver and one kidney sample.

## Example workflow

Fig 1 shows an example of the data acquisition and image analysis procedure for an ~1-cm$^2$ mouse kidney tissue sample (Fig 1A). Zoomed-in images are acquired for various fields of view (FOVs). Each image consists of a z-stack of seven z-positions, with a space of 0.7 $\mu m$ between each z-position. Each FOV is first imaged for DAPI and cell boundary antibody stains as shown in Fig 1B and C. Then, multiple rounds of hybridization begin and the various smFISH signals are acquired as shown in Fig 1D for a single bit.

From examining the presence, absence, and co-localization of individual smFISH spots, the digital barcodes of unique RNA species can be decoded using computational image analysis (Fig 1E, colored points; see the Materials and Methods section for details). The cell boundary antibody stain and DAPI channels can be used to segment and produce single-cell boundaries (Fig 1E, black). These result in processed images containing the single-cell boundaries and decoded transcript positions, which are then used for downstream bioinformatics analysis.

Sample quality and RNA integrity are critical to successful MERFISH analysis. We computed the average density of detected RNA transcripts for each tissue and compared this number with the RNA integrity number (RIN), a common metric for measuring RNA quality (Schroeder et al, 2006). The two metrics correlated quite well (see the Materials and Methods section and Fig S2), and we discarded measurements with an RIN score lower than four. All five liver datasets passed this margin, as well as two kidney datasets.

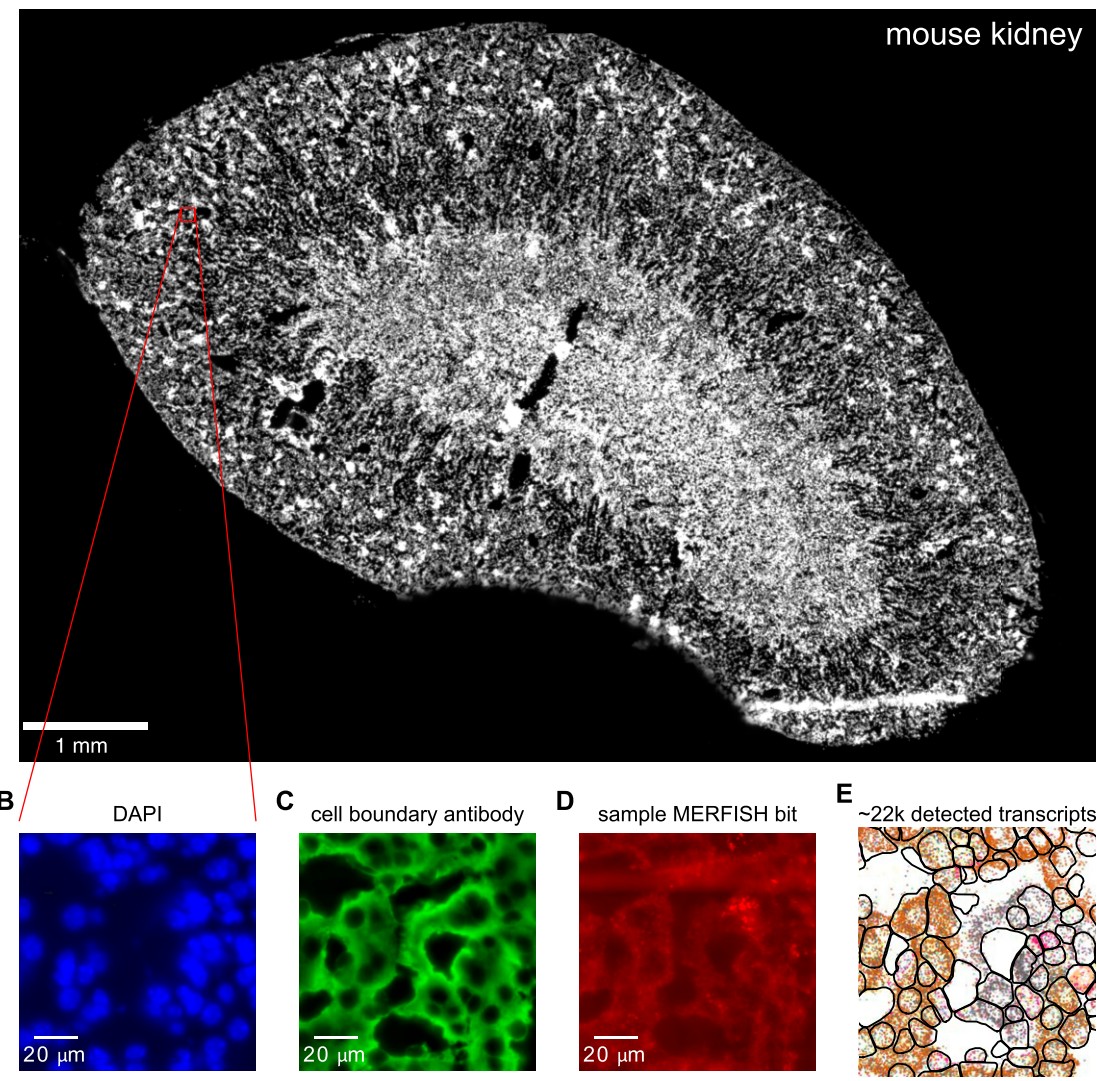

**Figure 1. Sample multiplexed error-robust fluorescence in situ hybridization data acquisition workflow.**
**(A)** Low-resolution image of the DAPI channel for a mouse kidney tissue sample. Red box indicates zoomed-in region displayed in B-E. **(B, C)** DAPI (B) and (C) cell boundary antibody stain channels for the zoomed-in region. **(D)** Multiplexed error-robust fluorescence in situ hybridization signal for a single barcode bit channel in the same zoomed-in region. **(E)** Positions of decoded mRNA transcripts (colorful dots) and segmented cell boundaries (black) in the same zoomed-in region after running data through image analysis pipeline, with each color representing a unique gene species. Panels (B, C, D) show raw images with brightness and contrast levels selected for ease of visualization.

## Comparison of MERFISH results with bulk RNA-seq

First, we examined noise in MERFISH measurements by comparing bulk RNA transcript counts between technical replicates. Fig 2A and B show total log-transformed counts of detected RNA transcripts for each gene in the panel between two technical replicates taken from the same tissue block in the mouse liver and kidney. Both tissues exhibited high correlation, with R = 0.99 and R = 0.95 for the liver and kidney, respectively. MERFISH replicates are extremely reproducible.

Then, we compared bulk MERFISH results with bulk RNA-seq data from Tabula Muris Senis (Schaum et al, 2020). Because the mice used in this study were 3 mo old, we only considered those mice from Tabula Muris Senis that were also 3 mo old. Fig 2C and D show the total log-transformed counts per gene in the mouse liver and kidney between the two types of technologies, with some marker genes highlighted for illustrative purposes. Each point represents the RNA count for a single gene, averaged across different experimental samples for the corresponding technology. Although the correlation was lower than between MERFISH technical replicates, it was still apparent (R = 0.61 and R = 0.58 for the liver and kidney, respectively), albeit slightly lower than previous MERFISH studies in cell culture (Xia et al, 2019). Furthermore, MERFISH systematically detected more transcripts than bulk RNA-seq, with fold-change increases of ~10× to over ~1,000×.

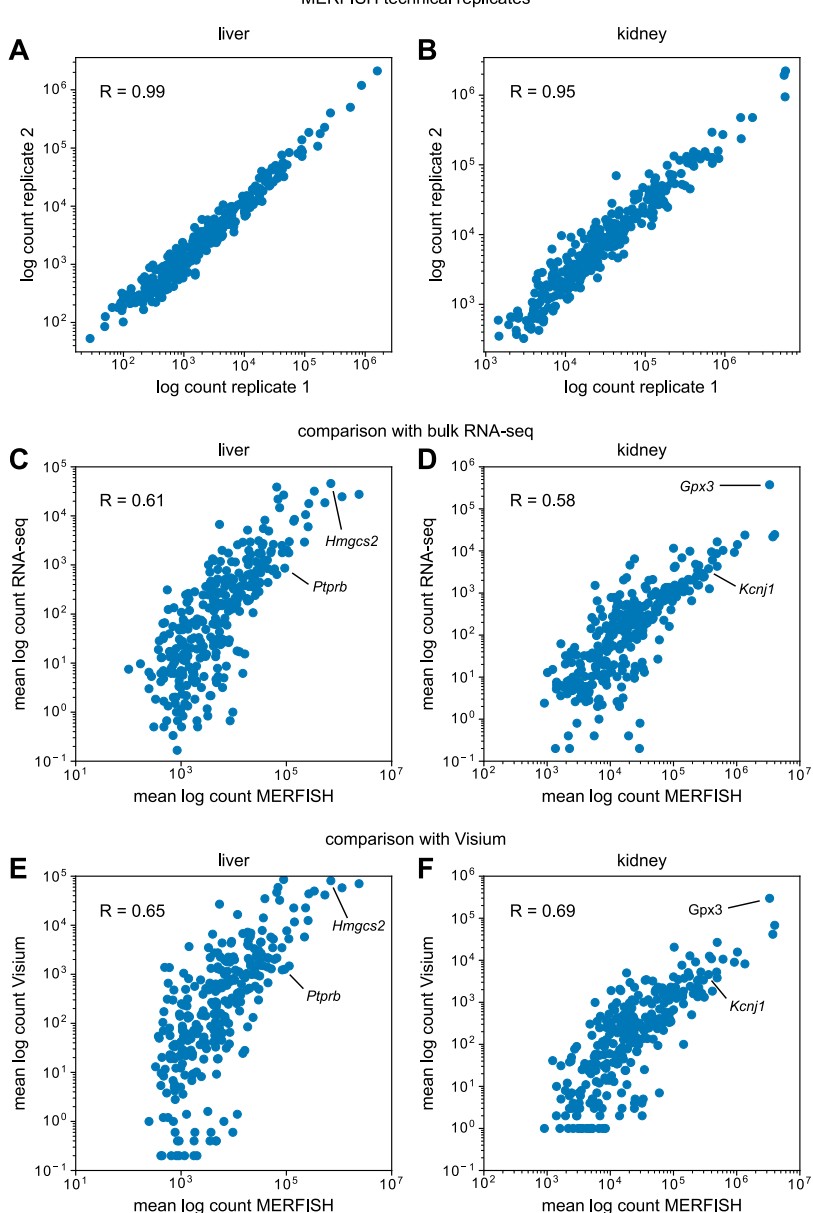

**Figure 2. Comparison of multiplexed error-robust fluorescence in situ hybridization (MERFISH) data with bulk RNA-seq from Tabula Muris Senis.**
**(A, B)** Bulk RNA counts per gene in the mouse (A) liver and (B) kidney between MERFISH sample replicates. **(C, D)** Comparison of bulk RNA counts per gene in the mouse (C) liver and (D) kidney between MERFISH and RNA-seq. **(E, F)** Comparison of bulk MERFISH RNA counts per gene with pseudo-bulk RNA counts from Visium in the mouse (E) liver and (F) kidney. In (C, D, E, F), counts per gene were averaged across replicates for each technology.

As an additional point of comparison, we examined bulk MERFISH results together with another spatial transcriptomics technology, the sequencing-based Visium assay (10x Genomics). We used two publicly available datasets for the mouse liver (Guilliams et al, 2022) and kidney (Dixon et al, 2022), respectively, and conducted the same correlation analysis as before. Because Visium is not a single-cell technology, we reasoned that this bulk analysis would be the best way to compare the two. For each Visium dataset, we created a pseudo-bulk measurement by simply summing up all of the detected RNA in the sample and then compared these data with the bulk MERFISH results.

Fig 2E and F show the results of this pseudo-bulk comparison for the mouse liver and kidney, respectively, with some marker genes highlighted. Like before, each point represents the RNA count for a single gene, averaged across different experimental samples for the corresponding technology. In both tissues, bulk MERFISH results correlated well with Visium (R = 0.65 and R = 0.69 for liver and kidney, respectively). Thus, at the bulk level, we conclude that MERFISH quantitatively agrees with both RNA-seq and Visium technologies.

## Analysis and quality control of single-cell MERFISH results

After image acquisition, we extracted 2D segmented cell boundaries and decoded RNA transcripts positions as shown for sample zoomed-in views at a single z-position in Fig 3A (liver) and Fig 3E (kidney). Using the segmented boundaries in the median z-position (i.e., 4th of 7), we assigned transcripts to cells by treating each cell's

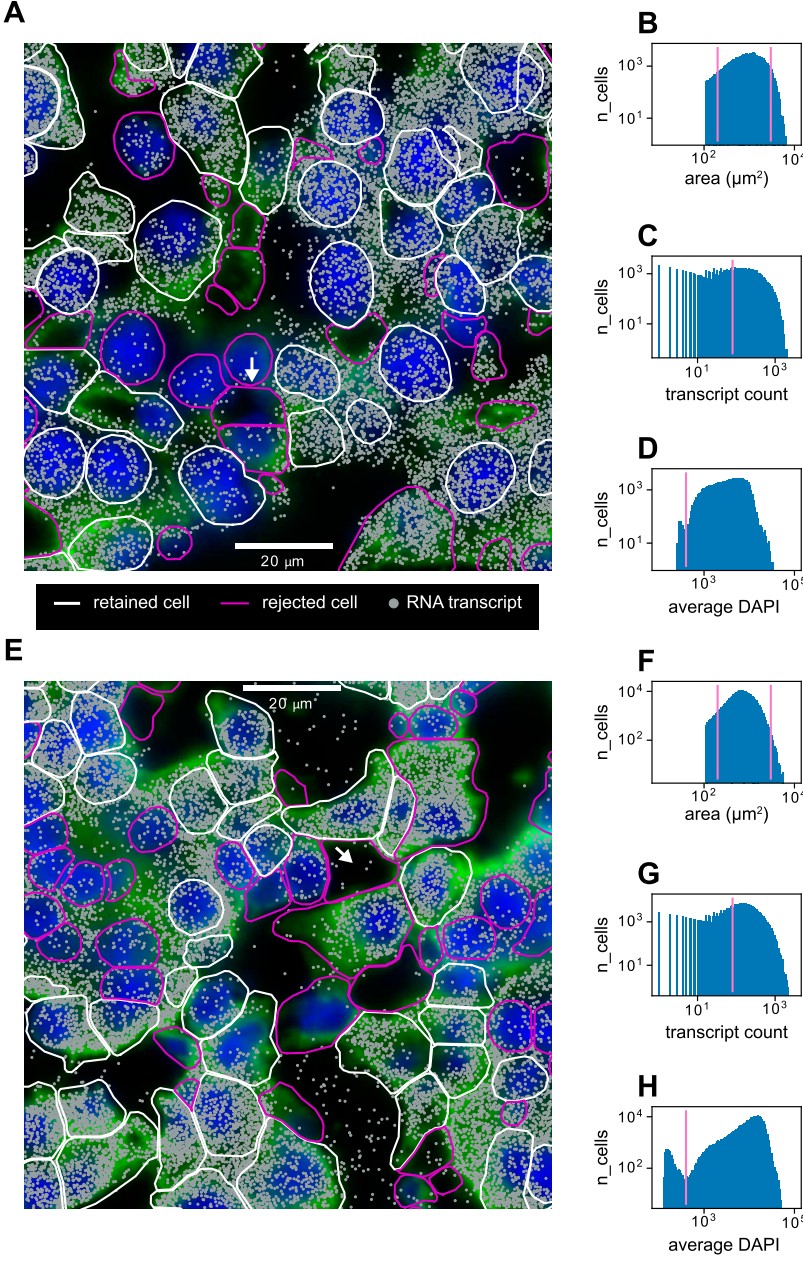

**Figure 3. Quality control of single-cell multiplexed error-robust fluorescence in situ hybridization data.**
**(A)** Sample cropped image of mouse liver tissue with segmented cell boundaries (white, magenta), decoded RNA transcripts (gray), DAPI nuclear stain (blue), and cell boundary antibody stain (green). White boundaries indicate cells that have passed the filtering stage, whereas magenta boundaries indicate cells that have been thrown out. **(B, C, D)** Histograms of (B) cell areas, (C) RNA transcript count per cell, and (D) average DAPI per cell for the image in (A). Magenta indicates minimum or maximum cutoff values for filtering criteria. **(E)** Sample cropped image of mouse kidney tissue. **(F, G, H)** Histograms of (F) cell areas, (G) RNA transcript count per cell, and (H) average DAPI per cell for the image in (E). White arrows in (A) and (E) indicate examples of poor cell segmentation.

2D boundary as a cookie cutter and associating every transcript in the whole 3D volume that lay within the boundary's xy coordinates with that cell. Segmentation quality was variable, with some cells possessing segmented boundaries that appeared quite reliable and others possessing aberrant segmented boundaries. These discrepancies were because of low-quality cell boundary and/or DAPI stain signals, resulting in the image analysis software constructing segmented boundaries in the empty spaces between cells (Fig 3A and E, white arrows). Not all detected RNA transcripts lay inside segmented boundaries—the percentage of transcripts assignable to cells was around 70% for both the liver and kidney. The remaining 30% likely consisted of both extracellular RNA and RNA

near the peripheries of cells that lay slightly outside the segmented boundaries.

We developed a quality control protocol to filter out these low-quality cells. From the data, we generated distributions of single-cell 2D areas and total counts per cell (Fig 3B, C, F, and G). We calculated a per-cell alignment metric with the DAPI signal which we designate as the *average DAPI* score. This score is the average DAPI intensity within the xy borders of each segmented cell in the median (i.e., fourth) z-slice (Fig 3D and H). A well-segmented cell would intuitively have a high-average DAPI score because of good alignment between the segmentation and the DAPI signal, whereas a poorly segmented cell would have a lower score.

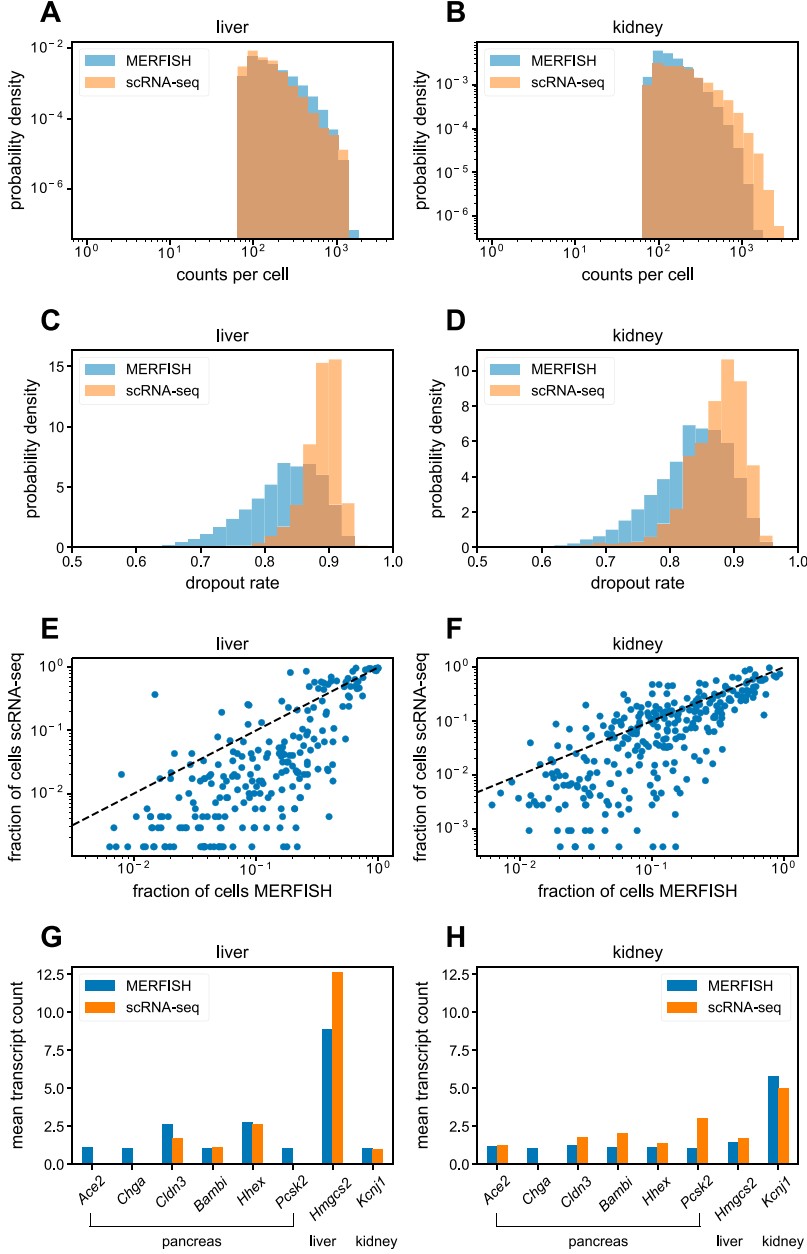

**Figure 4. Comparison of single-cell multiplexed error-robust fluorescence in situ hybridization (MERFISH) data with Tabula Muris Senis.**
**(A, B)** Histograms of total RNA transcript count per cell in the mouse (A) liver and (B) kidney for the gene panel, for MERFISH (blue) and scRNA-seq (orange). **(C, D)** Histograms of dropout rates per cell in the mouse (C) liver and (D) kidney for the gene panel. **(E, F)** Comparison between MERFISH and scRNA-seq of per-gene fraction of total cells of the whole population of each dataset that have nonzero counts for the mouse (E) liver and (F) kidney. Here, each dot represents a single gene. **(G, H)** Mean transcript count of pancreas marker genes in the mouse (G) liver and (H) kidney, for MERFISH (blue) and scRNA-seq (orange). A liver marker gene (*Hmgcs2*) and kidney marker gene (*Kcnj1*) are shown for comparison. Standard error of the mean across cells was negligible and too small to visualize.

Quality control filters were established as follows: segmented cells that were unreasonably small or large were discarded by establishing low-area and high-area cutoffs of 200 and 3,000 $\mu m^2$ in the distribution of cell areas (Fig 3B and F). A minimum cutoff of total RNA transcript count per cell of 80 was established to filter out cells with sparse statistics (Fig 3C and G). Both the mouse liver and kidney datasets exhibited bimodality in the median average DAPI score per cell (Fig 3D and H), and a cutoff of 400 was established to retain cells with a high score. This allowed us to filter out segmented cells that were actually composed of an empty space (Fig 3A and E, white arrows).

This quality control procedure resulted in 34,217 liver cells and 126,547 kidney cells being retained of 83,410 and 212,090 originally segmented cells, respectively. This corresponds to yields of 41% and 60% for the liver and kidney datasets, respectively.

## Comparison of single-cell MERFISH results with scRNA-seq

Using segmented data, we constructed single-cell RNA count matrices to compare the single-cell MERFISH results with the 3-mo Tabula Muris Senis scRNA-seq data (Tabula Muris Consortium, 2020). First, we examined the distribution of total RNA transcript counts per cell shown in Fig 4A and B for the mouse liver and kidney. For a proper comparison, the Tabula Muris Senis data were reduced to a subset containing the same 307 genes as the MERFISH gene panel. In the liver, the MERFISH distribution quantitatively agreed

with scRNA-seq (Fig 4A). In the kidney, the distributions agreed up to total counts of ~1,000 transcripts/cell, after which the MERFISH distribution drops to zero, whereas the scRNA-seq distribution continues.

We hypothesize that this is because of the fact that the gene panel contained a few highly abundant genes in the mouse liver and kidney such as *C1qc* and *Gpx3*, respectively (see the Materials and Methods section and Fig S3). For the minority of cells containing extremely high numbers of transcripts (i.e., over 1,000 per cell), the smFISH spots in the MERFISH images would be too dense because of overcrowding of RNA molecules, preventing accurate identification of single-RNA transcripts. Examination of single-cell distributions of these abundant genes verified this hypothesis as the distributions resulting from MERFISH agreed with those from scRNA-seq for cells with low overall transcript counts but not for cells with high overall transcript counts (see the Materials and Methods section and Fig S3). This effect was more pronounced in the kidney than in the liver, reflecting the increased discrepancy in overall RNA count distributions in the kidney (Fig 4B).

Next, we examined the dropout rate between the technologies, shown in Fig 4C and D. For each cell in a given tissue, this was defined as the fraction of genes with zero counts of the whole 307-gene panel. For both technologies, the dropout rates were high (>0.7) because of the fact that the gene panel contained marker genes specific for the liver, kidney, and pancreas. Nevertheless, the relative difference in the distribution between technologies was informative. In both the liver and kidney, although both technologies possessed cells with high dropout rates (>0.8), MERFISH possessed a larger fraction of cells with lower dropout rates (between 0.7 and 0.8) compared with scRNA-seq. Thus, we conclude that MERFISH results in systematically lower dropout rates than scRNA-seq.

We then investigated whether either technology was more sensitive than the other (Fig 4E and F). Although the previous analysis of dropout rates quantified the number of genes detected per cell, we now quantified the number of cells in which each gene was detected. To do so, we calculated the fraction of cells that had nonzero counts for each gene. If the two technologies had identical sensitivities, then a scatter plot of this fraction for each gene between technologies would fall on the x = y line. Bias in either direction would indicate that one technology was more sensitive. In both the liver (Fig 4E) and the kidney (Fig 4F), MERFISH systematically detected genes in higher proportions of cells than scRNA-seq. Thus, for this gene panel, MERFISH is more sensitive than scRNA-seq.

Finally, we investigated possible detection of false positives (i.e., incorrectly decoded RNA transcripts) in MERFISH. To do so, we reasoned that the pancreas marker genes in the panel could be used as an effective control—substantial detection of these pancreas genes in the MERFISH data would indicate a nontrivial false-positive rate. Fig 4G and H show the mean transcript count of six pancreas marker genes (*Ace2*, *Chga*, *Cldn3*, *Bambi*, *Hhex*, and *Pcsk2*) among cells with nonzero counts in both the MERFISH and Tabula Muris Senis datasets. The SEs of the mean were also calculated but were too small to visualize. For comparison, a liver and kidney marker gene (*Hmgcs2* and *Kcnj1*, respectively) are shown as well. The values for both MERFISH and scRNA-seq were similar for the

pancreas genes with a few exceptions likely because of expected dropouts in scRNA-seq (*Ace2*, *Chga*, and *Pcsk2*), indicating that both technologies have comparable false-positive rates. In addition, the expression levels of *Hmgcs2* were high in the liver and low in the kidney, and the expression levels of *Kcnj1* were low in the liver and high in the kidney, as expected. There was a slight discrepancy in the measured expression of the highly expressed *Hmgcs2* transcript in the liver between MERFISH and scRNA-seq, probably because of the molecular-crowding effect already discussed. Together, these data indicated that the effective false-positive rates were low in MERFISH and scRNA-seq.

To investigate the statistical properties of MERFISH data more deeply, we compared the mean RNA counts for each gene between MERFISH and scRNA-seq, across cells that possessed nonzero counts for that gene (Fig 5A and B for the liver and kidney, respectively). We only considered genes that had at least 50 cells with nonzero counts in each dataset, to filter out genes with ambiguous means because of low sample size. In the liver (Fig 5A), there was a decent concordance and correlation of the means between MERFISH and scRNA-seq around the x = y line (R = 0.61), whereas in the kidney (Fig 5B), the correlation was slightly worse (R = 0.48). Furthermore, in the kidney, there was less concordance—the scRNA-seq mean RNA counts were systematically higher across genes. We hypothesized that this was because of a combination of the overcrowding effect mentioned earlier, which would reduce the number of cells in MERFISH detected with high RNA counts, with the segmentation procedure performing worse in the kidney and potentially failing to account for RNA transcripts located further away from cell nuclei.

scRNA-seq RNA counts are well modeled by the negative binomial distribution (Grün et al, 2014; Kharchenko et al, 2014). Although this has also been demonstrated to hold for smFISH (So et al, 2011), similar analysis is more sparse for MERFISH, although recent results suggest that MERFISH statistics are also overdispersed and captured with the negative binomial distribution (Zhao et al, 2022). We investigated this by examining the mean–variance relationship of RNA counts for both MERFISH and scRNA-seq for the genes shown in Fig 5A and B, again only for cells with nonzero counts. We opted to only examine cells with nonzero counts because MERFISH and scRNA-seq possessed different dropout rates (Fig 4C and D). Fig 5C and D show this relationship for the liver and kidney, respectively. As a reference, the case where RNA counts follow a Poisson distribution (i.e., the variance equals the mean) is shown in the black dashed line. For both the scRNA-seq (orange dots) and MERFISH (blue dots) data, the variance is smaller than the mean for genes with low expression and higher than the mean for genes with high expression. Thus, both scRNA-seq and MERFISH possess overdispersed statistics for abundant genes. Although this is expected for scRNA-seq, these results are novel in the case of MERFISH.

This overdispersion is well captured with the negative binomial distribution—the insets in Fig 5C and D show the full distribution of RNA counts for the liver and kidney marker genes *Hmgcs2* and *Kcnj1*, respectively (blue bars), along with fits to the negative binomial distribution (red lines). In addition, the mean–variance relationship for both MERFISH and scRNA-seq are remarkably similar in both the liver and kidney. Together, the data indicate that

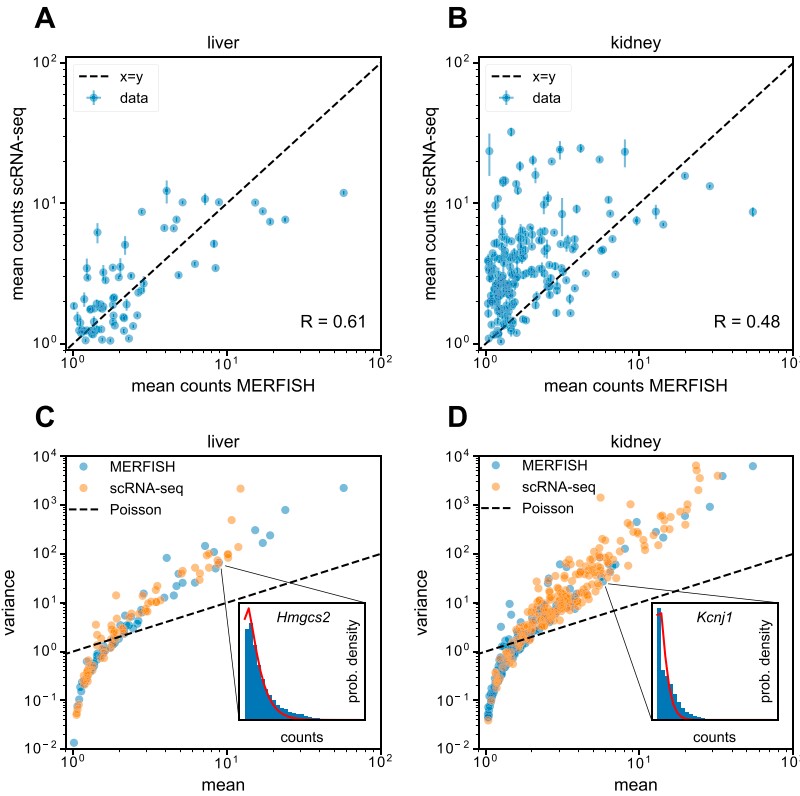

**Figure 5. Statistical analysis of single-cell multiplexed error-robust fluorescence in situ hybridization (MERFISH) and Tabula Muris Senis RNA count distributions.**
**(A, B)** Comparison of mean RNA counts in MERFISH versus scRNA-seq in mouse (A) liver and (B) kidney across cells with nonzero counts. Each dot represents a gene, and only genes possessing at least 50 cells with nonzero counts are shown for each tissue. The dotted black line shows the x = y line, indicating one-to-one concordance between MERFISH and scRNA-seq. **(C, D)** Mean–variance relationship for RNA counts in MERFISH (blue) and scRNA-seq (orange) for the mouse (A) liver and (B) kidney, across cells with nonzero counts. Each dot represents a gene, and only genes possessing at least 50 cells with nonzero counts are shown for each tissue. The dotted line indicates the x = y line, which represents the Poisson scenario where the mean equals the variance. (C, inset) Distribution of MERFISH RNA counts across cells with nonzero counts for *Hmgcs2*, a liver hepatocyte marker gene. The red line indicates the best fit to a negative binomial distribution. (D, inset) Distribution of MERFISH RNA counts across cells with nonzero counts for *Kcnj1*, a kidney loop-of-Henle cell marker gene. Red line indicates the best fit to a negative binomial distribution. Error bars in (A) and (B) represent the standard error of the mean.

for our gene panel, MERFISH and scRNA-seq RNA counts exhibit quantitatively similar RNA count statistics that are well modeled by a negative binomial distribution, for cells possessing nonzero counts.

## Single-cell and spatial analysis of MERFISH results

We explored MERFISH's ability to resolve distinct liver cell types on its own using the segmented single-cell results. First, we visualized a low-dimensional embedding using UMAP (McInnes et al, 2018) and conducted unsupervised clustering using the Leiden algorithm (Traag et al, 2019).

With just the 307-gene panel, we observed clear separation of clusters corresponding to various liver cell types in the UMAP plot (Fig 6A, left). By examining the most informative genes representative of each group, we were able to assign cell type identities to the various clusters. The three right plots in Fig 6A show expression levels of some marker genes for periportal hepatocytes (*Cyp2f2*), endothelial cells (*Ptprb*), and Kupffer (*Clec4f*) cells.

As expected, the bulk of the cells were hepatocytes. Notably, the data were able to resolve between different subpopulations of hepatocytes with both periportal and pericentral hepatocytes clearly represented in the data (Fig 6A, brown and pink). We were also able to distinguish between periportal and pericentral endothelial cells (Fig 6A, gray and olive). Finally, we detected a very small number of bile duct epithelial cells (Fig 6A, cyan; 60 cells of 34,217 total cells).

Projecting these cell type annotations onto the spatial plot of mouse liver allowed us to observe clear spatial structure of the different cell types (Fig 6B). In particular, the periportal and pericentral hepatocytes and endothelial cells exhibited spatial segregation via prominent maze-like patterning. Focusing on this zonation revealed clear co-localization of periportal hepatocytes and endothelial cells, as well as of pericentral hepatocytes and endothelial cells (Fig 6B, inset; see the Materials and Methods section; Fig S4). A DAPI stain of the sample is shown in Fig 6C for reference.

To see if we could further enhance our signal quality, we integrated the MERFISH liver dataset with the annotated Tabula Muris Senis liver cell atlas (Tabula Muris Consortium, 2020) using scVI (Lopez et al, 2018). To do so, we retained mice from the atlas that were 3 mo old, in addition to non-hepatocyte cells from the 1-mo dataset to increase cell diversity as the 3-mo data primarily consisted of hepatocytes (see the Materials and Methods section). Altogether, 2,321 liver cells from scRNA-seq were used. We then subsetted these data to the same 307-gene panel as the MERFISH data and then trained an scVI model to combine the two datasets. Details of the analysis are given in the Materials and Methods section, and the combined UMAP in the joint embedding is shown in Fig S5A. Finally, we used scANVI (Xu et al, 2021) to predict cell type labels in the MERFISH dataset from the annotations from Tabula Muris Senis (Fig S5B). The resulting spatial plot with projected scANVI cell type annotations (Fig 6D) qualitatively reproduced the spatial results from the MERFISH data alone (Fig 6B).

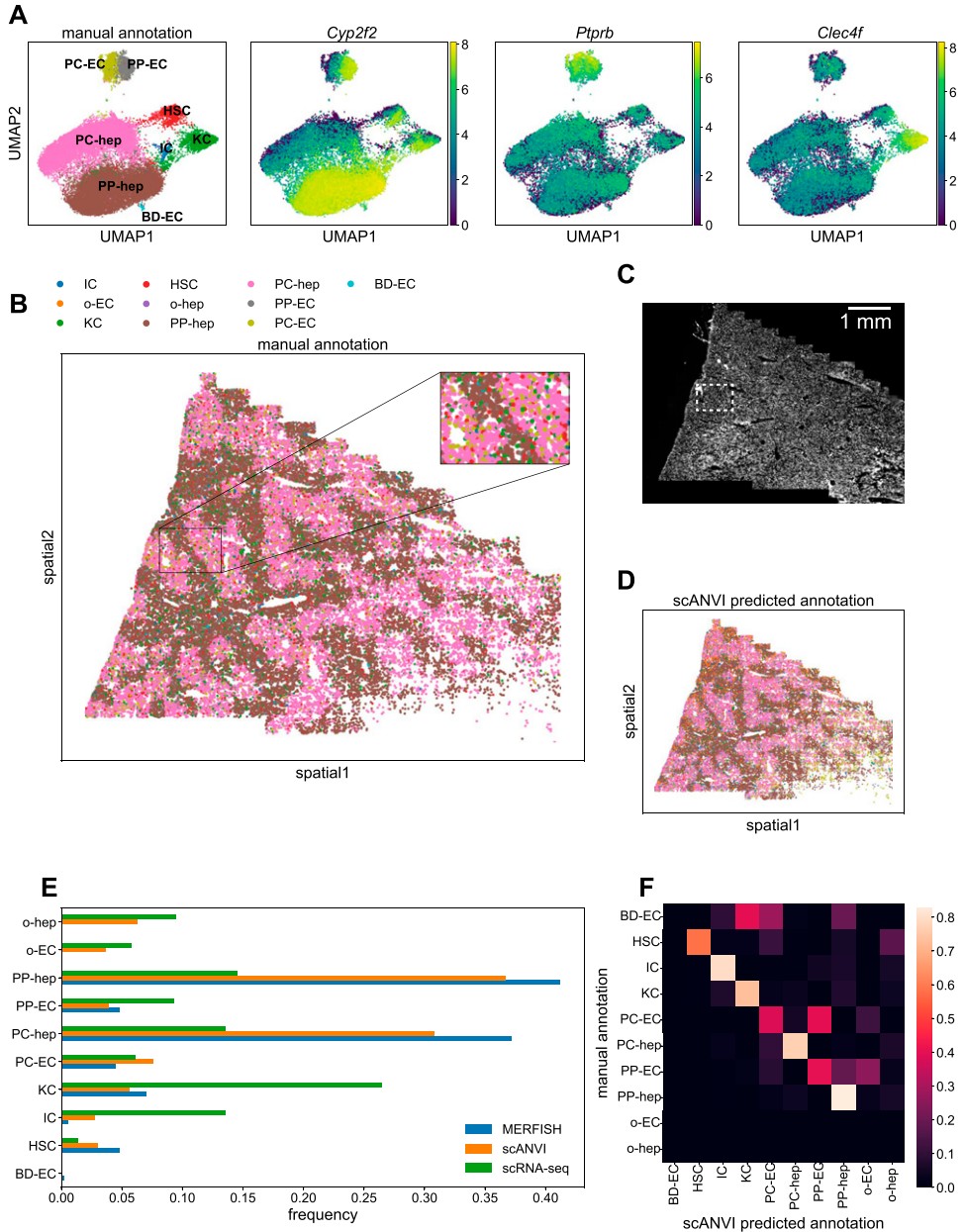

**Figure 6. Single-cell and spatial analysis of the multiplexed error-robust fluorescence in situ hybridization (MERFISH) liver sample.** **(A)** UMAP plots of MERFISH data colored by manually annotated clusters or by normalized, log-transformed, and scaled expression of example marker genes. **(B)** Spatial plot of MERFISH dataset alone, colored by manually annotated cell types in (A). Inset highlights spatial co-localization of periportal endothelial cells and hepatocytes (gray and brown, respectively) and pericentral endothelial cells and hepatocytes (olive and pink, respectively). **(C)** DAPI stain of the liver sample. White box indicates the same inset region as in panel (C). **(D)** Spatial plot of MERFISH dataset using scANVI predicted cell type labels. Legend is the same as in panel (B). **(E)** Cell type composition for scRNA-seq and MERFISH datasets. Each point in (A, B, D) represents a single cell. **(F)** Confusion matrix of MERFISH cell type annotations between the manual method and scANVI predictions. The rows for other endothelial cells and hepatocytes ("o-EC" and "o-hep") are blank because none of the manual annotations were in these groups. In addition, the row for bile duct epithelial cells is noisy because none of the scRNA-seq reference cells possessed this annotation. Cell type abbreviations are as follows: "IC," "immune cell"; "o-EC," "other endothelial cell"; "KC," "Kupffer cell"; "HSC," "hepatic stellate cell"; "o-hep," "other hepatocyte"; "PP-hep," "periportal hepatocyte"; "PC-hep," "pericentral hepatocyte"; "PP-EC," "periportal endothelial cell"; "PC-EC," "pericentral endothelial cell"; "BD-EC," "bile duct epithelial cell."

Comparisons of cell type frequencies between the scRNA-seq reference, MERFISH data, and scANVI integration are shown in Fig 6E. There was a discrepancy between scRNA-seq and MERFISH in the frequencies of hepatocyte subpopulations, likely because the tissue sample used here did not contain the whole liver and thus lacked a globally representative population of hepatocytes. The scRNA-seq reference contained other hepatocytes and endothelial cells that were neither periportal nor pericentral. These cell types were not readily detected in the manual annotation scheme used for the MERFISH data. More notably, MERFISH detected substantially fewer immune and Kupffer cells and more hepatic stellate cells than the scRNA-seq reference, hinting at the fact that in situ hybridization technologies could offer a more accurate estimate of

cell type proportions because scRNA-seq may overestimate immune cell counts (Wu et al, 2019; Denisenko et al, 2020; Ding et al, 2020; Koenitzer et al, 2020; Slyper et al, 2020). In addition, bile duct epithelial cells were not detected in the scRNA-seq reference and thus were not predicted by scANVI. An independent integration using Harmony and Symphony (Korsunsky et al, 2019; Kang et al, 2021) yielded similar results (see the Materials and Methods section and Fig S6).

To directly compare the manual annotations with the predicted annotations from scANVI, we calculated the confusion matrix of cell annotations between the two methods. Each entry $i, j$ in the matrix was defined as the fraction of cells manually annotated with label $i$ that were predicted with label $j$ in scANVI. Thus, perfect agreement

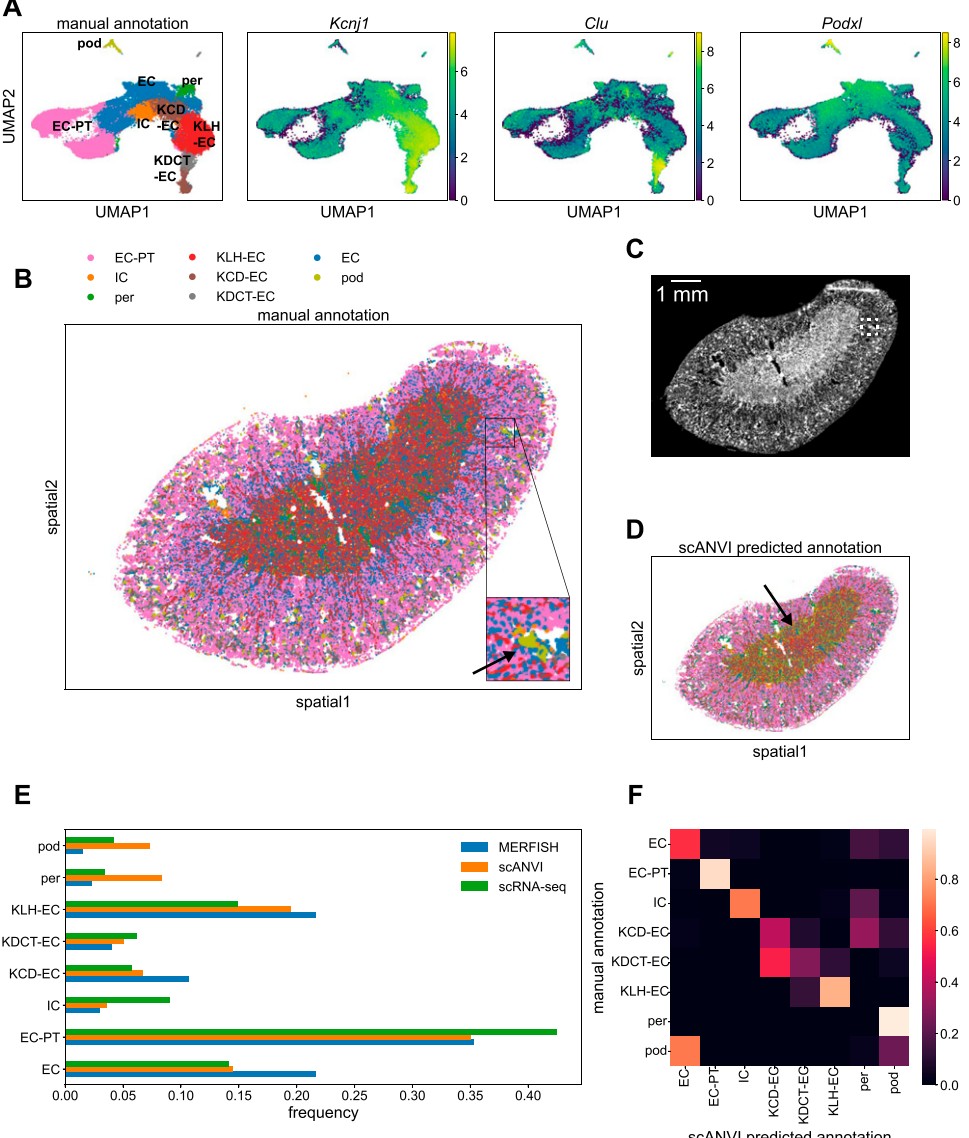

**Figure 7. Single-cell and spatial analysis of multiplexed error-robust fluorescence in situ hybridization (MERFISH) kidney sample.** **(A)** UMAP plots of MERFISH data colored by manually annotated clusters or by normalized, log-transformed, and scaled expression of example marker genes. **(B)** Spatial plot of MERFISH dataset alone, colored by manually annotated cell types in (A). Inset shows an example of a podocyte cluster in the kidney cortex region (olive, black arrow). **(C)** DAPI stain of kidney sample. White box indicates the same inset region as in panel (C). **(D)** Spatial plot of MERFISH dataset using scANVI predicted cell type labels. Black arrow indicates falsely predicted podocytes distributed in a ring-like structure around the medulla. Legend is the same as in panel (B). **(E)** Cell type composition for scRNA-seq and MERFISH datasets. Each point in (A, B, D) represents a single cell. **(F)** Confusion matrix of MERFISH cell type annotations between the manual method and scANVI predictions. Cell type abbreviations are as follows: "EC-PT," "epithelial cell of proximal tubule"; "IC," "immune cell"; "per," "pericyte"; "KLH-EC," "kidney loop-of-Henle epithelial cell"; "KCD-EC," "kidney collecting-duct epithelial cell"; "KDCT-EC," "kidney distal-convoluted-tubule epithelial cell"; "EC," "endothelial cell"; "pod," "podocyte."

would result in the diagonal elements possessing values of 1 and all other off-diagonal elements possessing values of zero. For the liver data, there was generally high agreement between the two methods (Fig 6F). Because bile duct epithelial cells were not present in the scRNA-seq reference, there was obviously no agreement between scANVI and the manual MERFISH annotations. In addition, the manual MERFISH annotations did not label any "other" endothelial cells or hepatocytes, so those rows are blank.

We then repeated this analysis for the MERFISH kidney data. The left panel in Fig 7A shows a UMAP plot of the MERFISH data alone from a single tissue sample, where we could resolve distinct clusters with the Leiden algorithm. Clusters were annotated as before by examining the most informative genes in each cluster. The three right plots in Fig 7A show the example marker genes *Kcnj1, Clu*, and *Podxl*, which along with other marker genes allowed us to detect loop-of-Henle epithelial cells, distal-convoluted-tubule

epithelial cells, and podocytes, respectively. Although the kidney contains many more known cell types than the liver, we were able to resolve broad categories of cell types, including endothelial cells, various epithelial cell groups, and podocytes. We did not detect fibroblasts via manual annotation; however, even in the scRNA-seq reference, they were extremely rare (<1% of cells).

Fig 7B shows the corresponding mouse kidney spatial plot, with cell type labels projected from this annotation process. We observed clear stratification of the kidney into different layers—epithelial proximal tubule cells were localized to the cortex (Fig 7B, pink); loop-of-Henle epithelial cells were localized to the medulla (Fig 7B, red). Endothelial cells were more uniformly distributed throughout the kidney sample (Fig 7B, blue). Notably, we also discovered clusters of podocytes (Fig 7B, olive), a cell type that only exists in glomeruli (Brunskill et al, 2011), multicellular structures that exist dispersed throughout the kidney cortex. A DAPI stain is shown for reference in Fig 7C.

We then again used scVI to integrate the kidney MERFISH data with the Tabula Muris Senis kidney scRNA-seq reference atlas. Because the reference scRNA-seq kidney data were more comprehensive than the liver data, we used only 3-mo-old mouse data. 2,327 kidney scRNA-seq cells were used in total. This resulted in the joint UMAP plot shown in Fig S5C.

Using scANVI, we then predicted cell types in the MERFISH data from the annotations in Tabula Muris Senis (Fig S5D). Although the spatial kidney plot with predicted scANVI cell type labels exhibited qualitatively similar spatial structure (Fig 7D) to the spatial results from MERFISH alone (Fig 7B), it did produce some clear artifacts. For example, podocytes only exist in spatial clusters concentrated in glomeruli in the cortex, but the integration predicted the existence of a large podocyte population that lay in the medulla (Fig 7D, olive).

A comparison of the relative cell type frequencies between the scRNA-seq reference, MERFISH results, and scANVI integration is shown in Fig 7E. The kidney sample here covered the entire organ unlike the liver sample in Fig 6; therefore, the agreement between MERFISH and scRNA-seq was much more quantitatively apparent (Fig 7E, blue and green). The major differences were that MERFISH detected more endothelial cells, loop-of-Henle epithelial cells, and collecting-duct epithelial cells than scRNA-seq, as well as fewer podocytes. The scANVI integration produced many more discrepancies (Fig 7E, orange), such as an abnormally high podocyte and pericyte count compared with scRNA-seq.

Examining the confusion matrix clarified these aberrations (Fig 7F). Although most cell type annotations agreed well between the manual and scANVI annotations, both pericytes and podocytes were poorly predicted by scANVI compared with the manual annotation. In particular, more podocytes were predicted by scANVI to be endothelial cells compared with those podocytes that were correctly labeled. Integration with Harmony and Symphony reproduced the scVI and scANVI results, including these discrepancies (Fig S6).

Fig S7 explores the basis of this mislabeling in further detail. To summarize, for certain cell types (e.g., podocytes), the MERFISH data are systematically different enough from the same cell types in the scRNA-seq reference that any integration method likely would fail. In other words, the integration performs well for matching cells with similar gene expression profiles together, but the two dataset types themselves are fundamentally different enough that the integration is not well posed. We refer the reader to see the Materials and Methods section for a more detailed description of this investigation and conclude that the MERFISH-alone manual annotation is more reliable than computational integration for these cell types.

## Discussion

MERFISH and other evolving spatial transcriptomic technologies offer a paradigm shift in transcriptomic analysis by combining subcellular spatial resolution with single-cell segmentation ability. Although MERFISH promises to extend single-cell transcriptomic analysis to the spatial domain, it is unclear as to what extent MERFISH measurements are quantitatively comparable to those from traditional RNA sequencing technologies. A technical comparison with existing RNA sequencing data is necessary to fully understand the similarities and differences between the two modalities. An important question is if MERFISH independently enables cell type identification or if it requires computational integration with more comprehensive reference scRNA-seq databases.

Here, we used MERFISH to measure the RNA profiles of 307 liver, kidney, or pancreas cell type marker genes in the mouse liver and kidney. By using a fixation and clearing protocol in conjunction with an automated microfluidics setup and fluorescence microscopy, we measured single-molecule positions of the various genes in large (~1-cm$^2$) sections of fresh-frozen tissue. Combining cell segmentation based on membrane antibody and DAPI stains, we obtained bulk and single-cell RNA counts (Fig 1) that were then compared with the preexisting Tabula Muris bulk and single-cell RNA datasets.

We assessed technical noise by comparing RNA counts between technical replicates of mouse liver and kidney samples and found extremely high correlation between them (Fig 2A and B). Bulk comparisons with Tabula Muris Senis and publicly available Visium spatial transcriptomics data indicated relatively good agreement (Fig 2C–F). MERFISH typically produced bulk RNA counts that were several orders of magnitude higher than RNA sequencing, an important point in its favor for the measurement of sparse genes.

We then investigated the single-cell RNA counts from our MERFISH measurements. We first developed a quality control protocol to filter out poor cells based on metrics involving spatial morphology, transcript counts, and alignment with DAPI nuclear fluorescence (Fig 3). From this perspective, imaging-based transcriptomics methods such as MERFISH allow for easily interpretable quality control as these metrics can be ultimately derived from raw images rather than from downstream quantities. The predominant factor influencing signal quality was cell segmentation. After quality control, about half of detected cells were filtered out from a combination of low transcript count, extremal segmented cell areas, and/or poor alignment with DAPI nuclear fluorescence. Although this conservative filtering strengthens downstream signal quality, it also poses the need for improved computational segmentation in the future to fully harness the massive amounts of data contained in the raw images. Recent methods that boost cell segmentation quality by using information from RNA transcript positions (Petukhov et al, 2021) or prior knowledge of cell type–specific gene expression (Littman et al, 2021) may further refine segmentation quality and allow for improved construction of single-cell RNA count matrices from subcellular MERFISH signals. Such advances are necessary, as not all cell types may be equally affected by quality control—for example, more oddly shaped cells such as fibroblasts may likely be more difficult to segment and thus could systematically be excluded during quality control. Improvements in protein antibody staining could also boost segmentation quality in this regard and enable more robust detection of diverse cell types.

The filtered single-cell MERFISH results were then compared with the Tabula Muris Senis scRNA-seq database (Fig 4). We examined summary statistics in the mouse liver and kidney, starting with the total RNA counts per cell (Fig 4A and B). For the most part, MERFISH quantitatively reproduced the statistics from scRNA-seq. However, for the kidney, MERFISH was unable to resolve a minority of cells that possessed about 1,000 or more total RNA counts from the 307-gene panel (Fig 4B).

We hypothesized that this was because of the presence of highly abundant genes resulting in crowding of RNA molecules, bringing the density of fluorescent spots to a level exceeding that of the diffraction limit of the microscope and thus obscuring the ability to resolve them. Because MERFISH uses a barcode labeling scheme, this would then reduce signal quality in many genes in the panel for these highly expressing cells. Our experience recommends that future MERFISH experiments avoid highly expressing genes in the gene panel to ameliorate this issue. Although the gene panel design implemented here included a filtering step based on overall bulk RNA measurements (see the Materials and Methods section for description), future selection criteria could incorporate total RNA counts from single-cell reference data as well (see the Materials and Methods section and Fig S3). We expect the dynamic range of MERFISH measurements in the mouse liver and kidney to be limited to cells that have under ~1,000 labeled transcripts overall.

We also investigated measurement sensitivity by quantifying the number of genes measured per cell and the number of cells that measured each gene. First, we calculated the dropout rate in the liver and kidney, defined as the fraction of genes in the panel possessing zero counts per single cell (Fig 4C and D). The distribution of dropout rates was shifted to lower values for both the liver and kidney compared with scRNA-seq, suggesting that MER-FISH was more sensitive than scRNA-seq in those tissues. We then quantified per-gene sensitivity by examining the fraction of cells that possessed nonzero counts for each gene in the 307-gene panel, for both MERFISH and scRNA-seq (Fig 4E and F). In both the liver and kidney, this fraction was systematically larger for MERFISH than for scRNA-seq, again indicating MERFISH's increased sensitivity. Taken together, these results suggest that in situ hybridization technologies like MERFISH can measure a more intrinsically correct view of RNA statistics than a dissociative technology such as scRNA-seq.

Next, we undertook an effective negative control by investigating several genes in the panel that were marker genes for pancreas cell types. Examining the mean transcript count for these genes revealed similarly low values for both MERFISH and scRNA-seq compared with expression levels for positive marker genes (Fig 4G and H), indicating that MERFISH possessed low false-positive detection rates at a similar level to scRNA-seq. Overall, we conclude that MERFISH possesses quantitatively similar overall statistical behavior to scRNA-seq, with minor discrepancies such as the dynamic range issue discussed above.

To more deeply investigate the single-cell statistics of MERFISH, we examined the concordance and correlation of mean RNA counts among cells with nonzero counts between MERFISH and scRNA-seq (Fig 5A and B). In the liver, the concordance and correlation were decently high, whereas in the kidney, the correlation was weaker and scRNA-seq produced systematically higher counts (likely because of a combination of cell segmentation flaws and the overcrowding effect mentioned earlier). Nevertheless, this quantitative correspondence prompted us to investigate if MERFISH RNA counts could be modeled with similar statistics as scRNA-seq. Examining the mean–variance relationship of the genes studied in Fig 5A and B indicated that among cells with nonzero counts, MERFISH and scRNA-seq indeed possessed similar statistics (Fig 5C and D). Furthermore, like scRNA-seq, MERFISH RNA count distributions

were well modeled by a negative binomial distribution (Fig 5C and D, inset). Such analysis sheds further light on questions of noise in transcriptomic measurements (Grün et al, 2014) as the statistical agreement between MERFISh and scRNA-seq suggest that distributions such as the negative binomial actually capture true biological variability in cellular transcription rather than technical noise because of the measurement error.

Finally, we investigated MERFISH's ability to resolve distinct cell types in the mouse liver and kidney and if computational integration of MERFISH data with a scRNA-seq reference atlas could boost this resolving potential. Single-cell analysis of the MERFISH data using UMAP and Leiden clustering indicated reasonably good separation of cell types in both liver and kidney (Figs 6A and 7A). These unsupervised clustering results were clean enough to enable manual annotation of cell types based on representation of marker genes in the panel. After projecting these single-cell annotations onto a spatial plot, we were able to resolve clear spatial structure in both mouse liver and kidney samples (Figs 6B and 7B). The MERFISH data were able to distinguish between different liver hepatocyte and endothelial cell populations, exemplified by different spatial patterning of periportal and pericentral hepatocytes and endothelial cells (Fig 6B). Notably, although canonical marker genes for pericentral and periportal endothelial cells such as *Wnt2* and *Dll4* were not present in our gene panel, our MERFISH results could still identify these cells mainly on the basis of their spatial co-localization with the pericentral and periportal hepatocytes (Fig 6B, inset; see the Materials and Methods section; Fig S4). Thus, the incorporation of spatial information into analysis can greatly aid in annotation of cell types by comparing the annotations with expected spatial patterning.

Interestingly, computational integration of the MERFISH data with the Tabula Muris Senis scRNA-seq atlas using scVI did not noticeably improve cell type identification ability. The joint UMAP plots for both the liver and kidney were fairly noisy, especially for the kidney (Fig S5A and C), suggesting that MERFISH and scRNA-seq exhibit systematically different statistics at the individual RNA species level that hinder efficient harmonization. An additional complicating factor is in the throughput of cells as the individual MERFISH datasets used here possessed tens of thousands of cells, whereas the Tabula Muris Senis possessed only a few thousand total cells for each organ. Therefore, future investigations of this sort of computational integration should include a nuanced accounting of the imbalance in throughput between different technological modalities.

After using scANVI to predict cell type labels in the MERFISH data from the annotations in the Tabula Muris Senis dataset, we generated spatial plots of the mouse liver and kidney with these automatically generated labels (Figs 6D and 7D). Although the spatial patterning in the liver was qualitatively similar to the results from the manual annotation, the spatial results of the scANVI integration in the kidney produced a few artifacts. Direct comparison of cell type annotations between the manual method and scANVI indicated quantitative agreement for the liver (Fig 6F) and semi-quantitative agreement in the kidney (Fig 7F) with some exceptions for cell types such as podocytes. These results and discrepancies were reproduced by integration using Harmony and Symphony (Fig S6), suggesting that the underlying cause of the annotation

disagreement lay in the data rather than the integration method. Indeed, a deeper investigation into cell type RNA profile similarities and differences between MERFISH and scRNA-seq indicated that this was indeed the case (Fig S7). Thus, although computational integration works for the most part, we caution researchers to also annotate MERFISH data manually and check for potential discrepancies between annotation methods.

To more quantitatively investigate MERFISH's cell type resolving ability with and without computational integration, we compared cell type proportions between Tabula Muris Senis, MERFISH, and the integrated dataset in both the mouse liver and kidney (Figs 6E and 7E). Especially for the kidney, MERFISH resulted in cell type proportions that were generally closer to Tabula Muris Senis than the integrated dataset, further highlighting MERFISH's potential as a standalone technology. Interestingly, MERFISH detected far fewer immune cells than scRNA-seq in both organs (Figs 6E and 7E), supporting the hypothesis that scRNA-seq might overestimate immune cell counts and thus may not produce the most accurate reflection of minority cell types (Wu et al, 2019; Denisenko et al, 2020; Ding et al, 2020; Koenitzer et al, 2020; Slyper et al, 2020). This was further highlighted by the fact that MERFISH detected more hepatic stellate cells in mouse liver compared with scRNA-seq (Fig 6E). In combination with the increased measurement sensitivity discussed above (Fig 4C–F), our results anchor MERFISH as a technique that offers equivalent, if not better, overall biological signal compared with scRNA-seq. MERFISH and other spatial technologies provide a more complete representation of tissue cell types as fragile and rare cells may be lost in conventional tissue dissociation protocols.

We conclude that with efficient gene panel design, MERFISH cleanly resolves cell types and spatial structure alone and that computational integration with scRNA-seq reference datasets provide similar signal with varying quality depending on cell type. Although here we opted to use scVI and scANVI as well as Harmony and Symphony, many other computational methods exist, such as Seurat (Stuart et al, 2019), Tangram (Biancalani et al, 2021), and Giotto (Dries et al, 2021) to name a few examples. Further work should be undertaken to systematically compare these various integration methods for different tissue types and spatial measurement technologies, especially in light of the discovered systematic differences between MERFISH and scRNA-seq (Fig S7).

Substantial care should be taken regarding the design and interpretation of the gene panel. Because of the use of a scRNA-seq reference in creating the list of genes for the panel, the strength of MERFISH is heavily reliant on the quality of the reference and the method of panel construction. For example, the reference scRNA-seq dataset should be of sufficient sequencing depth to avoid creating a gene list based on noisy data. The method of choosing marker genes should also be robust. Although here we used simple differential expression analysis for picking marker genes, several recent software packages have been released that enable more nuanced approaches for marker gene panel creation (Aevermann et al, 2021; Missarova et al, 2021; Chen et al, 2022).

Importantly, because of the bespoke nature of these panels, MERFISH data are best examined in comparison to their parent scRNA-seq references. Comparing with other scRNA-seq datasets should be taken with caution because of the strong batch effects between various scRNA-seq datasets. In addition, MERFISH data will only be informative for the genes existing in the panel. Here, we used a panel consisting only of marker genes, so investigations of other biological effects such as cell state or metabolic activity are not possible. For future MERFISH experiments interested in such properties, relevant genes for study should be added to the list of marker genes already present in the panel. Because of the limited size of MERFISH gene panels, there hence will always be a constraint on the number of genes available for study, and efficient panel design is of paramount importance. For example, a follow-up study for the experiments here could use a single panel for each tissue of interest, freeing up space on the panels for non-marker genes relevant for effects such as cell cycle or state.

In light of these findings, we envision MERFISH primarily as a targeted approach for following up on preexisting scRNA-seq studies. Because the overall statistics of MERFISH measurements are quantitatively comparable to those from scRNA-seq, and because MERFISH data alone appear sufficient to robustly identify cell types and reproduce spatial structure, there appears to be no urgent need for computational integration between the two modalities, at least in the context of cell type identification in the liver and kidney.

# Materials and Methods

## Mouse tissue sample

The mice in this study were C57Black6 females and harvested at the age of 3 mo. The tissues of interest were fresh-frozen, embedded in an optimal cutting temperature compound, and stored at −80°C until cryosectioning. Tissues were collected by Patrick Neuhoefer in the Stanford Cancer Institute, Stanford Medical School. All animal care and procedures were carried out in accordance with institutional guidelines approved by the Stanford Medical School Committee on Animal Research.

## RNA quality measurement

To investigate the relationship between MERFISH signal quality and RNA integrity in the tissue, we measured the RNA integrity number (RIN) (Schroeder et al, 2006) in each of our tissue samples. The QIAGEN RNeasy Mini Kit was used to isolate RNA from tissue samples, and the RIN score was measured for each using the Agilent TapeStation system. As a metric for MERFISH signal quality, we calculated the *transcript density*, defined as the total number of detected RNA transcripts divided by the number of FOVs imaged using the microscope. The results of this investigation are shown in Fig S2.

## Design of gene panel

The MERFISH gene panel consisted of 307 genes, requiring 22 bits per barcode and eight rounds of three-color hybridization. The gene panel was designed by selecting the top 10 differentially expressed genes in each cell type according to Tabula Muris Senis cell atlas scRNA-seq data (Tabula Muris Consortium, 2020). Initially,

the list included 424 genes across 19 kidney, 19 liver, and 11 pancreas cell types. Genes with fewer than 30 target regions per transcript or higher than an abundance threshold of 800 FPKM (Fragments Per Kilobase of transcript per Million mapped reads) were removed from the list. The final 307 targetable genes possessed a total abundance of ~9,000 FPKM.

### Cryosectioning, staining, and hybridization

Frozen optimal cutting temperature–embedded tissue was sectioned at −15°C to a thickness of 10 μm and mounted onto a functionalized 20-mm coverslip treated with yellow green (YG) fluorescent microspheres. The mounted tissues were then fixed with 4% PFA in 1× PBS, washed with 1× PBS, and stored in 70% ethanol at 4°C for at least 1 d and no more than 1 mo before proceeding. Next, the cell boundary was stained using the Vizgen Cell boundary Staining Kit (Cat. no.: 10400009), a protein-based staining reagent consisting of a Blocking Buffer Premix (PN: 20300012), a Primary Staining Mix (PN: 20300010, 0.5 mg/ml), and a Secondary Staining Mix (PN: 20300011, 0.33 mg/ml) that can label 3 plasma membrane proteins in the cell. To perform the cell boundary staining, samples are first incubated with blocking solution supplemented with an RNase inhibitor for 1 h at room temperature. Then the sample is incubated with Primary Staining Solution diluted in Blocking Buffer (1:100 dilution) at room temperature for 1 h. The sample is washed three times with 1× PBS, 5 min each time, and then incubated with Secondary Staining Solution diluted in Blocking Buffer (1:33 dilution) at room temperature for 1 h. After washing with 1× PBS three times, 5 min each time, the sample is post fixed with 4% PFA at room temperature for 15 min and washed with 1× PBS. The sample was then treated with 30% formamide in 2× SSC (saline sodium citrate, formamide buffer) before the encoding probe hybridization buffer mix (MERFISH library mix) was applied. The sample was incubated for 36–48 h in a 37°C cell culture incubator while submerged in the MERFISH library mix.

### Gel embedding and tissue clearing

After hybridization of encoding probes to mRNA transcripts was complete, the sample was washed twice with a formamide buffer and embedded in an acrylamide/bis-acrylamide gel. Polymerization catalyzed by ammonium persulfate and NNN'tetramethylethylenediamine (TEMED) took ~1.5 h. Once gel formation was confirmed, the sample was rinsed briefly with 2× SSC and then incubated at 37°C in proteinase K–supplemented clearing solution (2% SDS, 0.5% Triton-X 100 in 2× SSC). After 1 d, the tissue became transparent and was ready for imaging.

### Imaging

The gel-embedded and cleared sample was washed repeatedly with 2× SSC to reduce autofluorescence from residual SDS in the gel. Then the first set of fluorescent probes (hybridization buffer A1) was pre-hybridized to the cell boundary. After 15 min, the sample was incubated in Wash Buffer from the Vizgen Imaging Reagent Kit for 10 min and gently assembled into the flow cell (Bioptechs, FCS2).

The reagents (Wash buffer, Imaging Buffer, Rinse Buffer, and Extinguishing Buffer) and hybridization buffers (up to 8) were loaded into the fluidic system, which was controlled by a user interface on a desktop computer. After priming the system, the flow cell was inserted into the fluidic path and Imaging Buffer was delivered to the sample.

Using a Nikon Eclipse Ti2 inverted microscope equipped with a 10× objective and 405-nm laser channel, a low-resolution mosaic was constructed from the resulting DAPI signal. Then the region of interest was selected, generating a text file containing a list of position coordinates. This position list, along with the corresponding fluidics recipe configuration file, was inputted to the automated fluidics and imaging control program. Using a 60× oil immersion objective, eight rounds of three-color imaging were performed. The cell boundary and DAPI stains were imaged at seven focal planes on the z axis for each tiled FOV. Imaging was followed by incubation in Extinguishing Buffer, Rinse Buffer, Hybridization Buffer (corresponding to the subsequent round of MERFISH readout probes), Wash Buffer, and Imaging Buffer. In each round, fluorescent probes were imaged at seven focal planes on the z axis using 749-, 638-, and 546-nm laser illumination. In addition, a single image of the fiducial beads was acquired at each FOV using 477-nm illumination. The resulting raw images stack was saved in various.dax files. As a point of reference, the raw images from a single run of MERFISH for a ~1-cm$^2$ tissue sample contain about 1 TB of data.

### Image analysis

To process the raw image files from the MERFISH experiments, we used the MERlin image analysis pipeline (Emanuel et al, 2020). Here, we briefly describe the pipeline. Initially, image stacks obtained from different MERFISH rounds are aligned to correct for microscope stage drift by maximizing their cross-correlation with fiducial bead images. The aligned images are then passed through a high-pass filter to remove background noise and a deconvolution process to clarify the RNA spots in preparation for bit-calling, that is, decoding a bit as 1 or 0 based on fluorescence detection (Guo et al, 2019 Preprint).

Individual RNA molecule barcodes are then decoded from using a pixel-based decoding algorithm that uses filters based on the area and intensity. Then, an adaptive barcoding scheme is used to correct misidentified barcodes that do not correspond to any barcodes in the codebook. The level of correction can be set to a user-specified final misidentification rate. Here, we set the misidentification rate to 5%.

Then, cells are segmented by using the information from the nuclear DAPI and cell membrane antibody stains with the cellpose software package (Stringer et al, 2021). The decoded RNA molecules are partitioned into individual cells to generate single-cell RNA count matrices. To do so, the xy segmentation from the median z-slice was used (the 4th position out of 7). After projecting this segmentation up and down through all z-positions, all detected transcripts in all z-positions that lay within the xy boundaries of a cell were assigned to that cell. Note that the whole pipeline is run for each imaging field of view and then tiled over the entire sample imaging area of ~1 cm$^2$.

## Investigation of molecular crowding in MERFISH measurements

To assess the impact of molecular crowding of RNA transcripts obscuring fluorescent signal, we reasoned that such an effect would be most apparent in the most abundant genes in the gene panel. Fig S3A and B show distributions of the median count of each gene in the MERFISH panel, calculated from cells in scRNA-seq that registered nonzero counts for that gene. Although most genes possess median counts fewer than 5, there are some genes with median counts over 10. The top two most abundant genes in the MERFISH panel from scRNA-seq were *C1qc* and *C1qa* in the mouse liver and *Gpx3 and Tmem27* in the mouse kidney.

We then hypothesized that molecular crowding should only occur in the regime of high transcript count. Thus, for cells with high overall numbers of transcripts, MERFISH measurements may become unreliable because of the high density of fluorescent spots making reliable single-molecule detection difficult. In contrast, for cells with low overall numbers of transcripts, MERFISH measurements should be more reliable.

To investigate this hypothesis, we examined the distribution of detected counts per cell for these two most abundant genes in the mouse liver and kidney, between MERFISH and scRNA-seq. Fig S3C and D show the results, where we split the analysis by overall transcript count. The top row shows these distributions for cells with total transcript count over 100, whereas the bottom row shows these distributions for cells with total transcript count under 100. In both the liver and kidney, we notice that the MERFISH and scRNA-seq distributions differ substantially for higher count values in cells with over 100 total transcripts. In contrast, the distributions are more similar in cells with under 100 total transcripts, particularly for the kidney.

Thus, we concluded that our MERFISH measurements were entering the molecular crowding regime, causing the quantitative agreement in statistics between MERFISH and scRNA-seq to diverge for cells with high overall RNA count number.

## Quantifying co-localization of periportal and pericentral endothelial cells and hepatocytes in the mouse liver

To quantify the co-localization of periportal and pericentral endothelial cells and hepatocytes (Fig 6B inset), we computed the co-occurrence probability likelihood of finding each cell type. To do so, we calculated the cluster co-occurrence ratio *R* provided in the *squidpy* package with the *squidpy.gr.co_occurrence* function (Palla et al, 2022). Briefly, the ratio describes the probability of locating a cell type *i* in a radius of size *d*, conditioned on the existence of a cell type *j*:

$$R(d) = \frac{P(i|j, d)}{P(i, d)}$$

The ratio is computed at varying length scales *d*. Thus, higher values of *R* correspond to higher likelihoods of co-localization and lower values to lower likelihoods of co-localization. In the case of spatial independence of two cell types, the ratio then takes a value of one. The co-occurrence probability likelihood results are shown in Fig S4.

## Comparison with Tabula Muris Senis reference cell atlas

Comparisons of bulk and single-cell MERFISH results were done with the bulk and droplet-based single-cell RNA-seq results in Tabula Muris Senis (Schaum et al, 2020; Tabula Muris Consortium, 2020). To remove the effects of age, we subsetted both reference datasets to mice that were 3 mo old.

## Comparison with publicly available Visium dataset

Comparisons of pseudo-bulk counts between Visium and MERFISH results were done with publicly available Visium datasets for the mouse liver and kidney (Dixon et al, 2022; Guilliams et al, 2022). The pseudo-bulk counts computed per each dataset were averaged over replicates for five Visium datasets for the mouse liver. For the mouse kidney Visium dataset, we used one sham dataset that Dixon et al (2022) used in their study as a control (healthy mouse).

## Single-cell bioinformatics analysis

Single-cell MERFISH and RNA-seq results were analyzed using *scanpy* (Wolf et al, 2018). All results were preprocessed by filtering out cells with low counts, extremal segmented areas, and low average DAPI scores as mentioned in the main text (Fig 3). The comparative analyses in Figs 4 and 5 used the raw counts that resulted from the preprocessing. For the single-cell analysis in Figs 6 and 7, the data were further normalized to a total count of 10,000 transcripts per cell, log-transformed, and scaled such that each gene possessed zero mean and unit variance across cells.

For cell type identification and annotation of MERFISH results alone, principal components were first computed using *scanpy*'s *tl.pca()* function with default settings. Then, neighborhood graphs were computed using *scanpy*'s *pp.neighbors()* function. UMAP plots and Leiden clustering were calculated using *tl.umap()* and *tl.leiden()*. To annotate the computed clusters, we examined the top differentially expressed genes in each cluster using the *tl.rank_genes_groups()* function and compared with known marker genes for the various cell types.

## Integration of MERFISH data with scRNA-seq reference using scVI and scANVI

For integrated analysis with scVI and scANVI, the Tabula Muris Senis scRNA-seq data were subsetted to retain the same 307 genes as the MERFISH data. For the mouse liver, the 3-mo data did not possess many non-hepatocyte cells, so we included non-hepatocyte data from the 1-mo data to increase cell diversity. For the mouse kidney, we used only 3-mo data. The MERFISH and scRNA-seq datasets were concatenated, and then we trained a joint model using scVI's default settings. The resulting neighborhood graph and UMAP plots were computed on the scVI latent space. For cell type annotation transfer, scANVI was run using 20 epochs and 100 samples per label.

The liver cell type annotations in Tabula Muris Senis were revised after consultation with tissue experts. Furthermore, the kidney cell type annotations were overly specific, so we coarse-grained them into broader categories.

### Integration of MERFISH data with scRNA-seq reference using Harmony and Symphony

To integrate MERFISH data with Tabula Muris Senis using Harmony and Symphony, we subsetted the scRNA-seq first using the same method mentioned above ("Integration of MERFISH data with scRNA-seq reference using scVI and scANVI). We then used Harmony and Symphony on the subsetted scRNA-seq to build a reference using the function *symphony::buildReference* with the following settings: 100 Harmony clusters, 100 variable genes to choose per group, and 20 principal components. The cell type annotations of MERFISH data were then predicted using the functions *mapQuery* with default settings and *knnPredict* with k = 5 neighbors.

### Investigation of systematic differences between MERFISH and scRNA-seq data

The discrepancies produced by both scANVI and Symphony integration for cell types such as podocytes led us to investigate if the integration methods were at fault or if the issue lay within the data itself. To do so, we strove to quantify the similarity in RNA count profiles between different cell types to see if systematic differences between MERFISH and scRNA-seq were a possible source for the integration errors.

We reasoned that perhaps MERFISH RNA count profiles for a certain cell type were more similar to a different cell type in the scRNA-seq reference than for the same cell type. For example, in our kidney data, both integration methods consistently mis-classified podocytes as endothelial cells (Figs 7F and S7D). To quantify the source of this error, we used the cosine similarity to compare the similarity of MERFISH podocytes with scRNA-seq podocytes and endothelial cells. Specifically, for each MERFISH podocyte, we calculated a mean cosine similarity value, defined thusly. For a given cell type pair (e.g., MERFISH podocyte versus scRNA-seq podocyte), we calculated the cosine similarity between each individual pair of MERFISH/scRNA-seq cells. We then averaged across scRNA-seq cells, generating a mean cosine similarity value for each MERFISH cell and a distribution of mean cosine similarity values across MERFISH cells.

Fig S7A shows the distribution of mean cosine similarity values between MERFISH kidney podocytes and scRNA-seq kidney podocytes (blue) or scRNA-seq kidney endothelial cells (orange), calculated by using the cosine similarity across all 307 genes in the panel. Surprisingly, the distribution for kidney endothelial cells possesses larger values than for podocytes, suggesting that MERFISH podocytes may actually have more similar gene expression with scRNA-seq kidney endothelial cells. Even if we compute the mean cosine similarity using just the podocyte marker genes in the panel (*Wt1*, *Actn4*, *Synpo*, *Dag1*, *Foxc1*, *Podxl*, and *Mme*), the MERFISH podocytes appear just as similar to scRNA-seq kidney endothelial cells as to scRNA-seq podocytes (Fig S7C). We note that this may stem from technical errors such as mis-segmentation of MERFISH podocytes or inaccurate annotation in the reference; nevertheless, we conclude that this is the main reason for the inaccurate cell type annotation prediction from integration.

As a control, we then compared mean cosine similarity values between MERFISH liver periportal hepatocytes and scRNA-seq liver periportal hepatocytes or pericentral hepatocytes (Fig S7B). These were cells that were very accurately classified with the integration methods (Figs 6F and S6C). After considering only marker genes (Fig S7D), the MERFISH liver periportal hepatocytes were much more similar to scRNA-seq liver periportal hepatocytes than to scRNA-seq liver pericentral hepatocytes.

## Data Availability

Raw MERFISH data are available for download on AWS s3 at https://registry.opendata.aws/czb-tabula-muris-senis/spatial-transcriptomics/MERFISH-data/. The gene panel codebook, RIN scores, FISH probe target region sequences, and processed data including decoded MERFISH transcript information, cell boundaries, bulk RNA statistics, and single-cell RNA counts with cell type annotations are available for download from figshare at https://figshare.com/projects/MERFISH_mouse_comparison_study/134213 (Pisco, 2022a, 2022b, 2022c, 2022d, 2022e, 2022f, 2022g, 2022h, 2022i). A Github repository containing the code needed to reproduce the figures is available at https://github.com/czbiohub/MERFISH-mouse-comparison.

## Supplementary Information

## Acknowledgements

We thank Patrick Neuhoefer for providing the frozen OCT embedded mouse tissue samples used in this work and Gabriel Loeb for helpful discussions regarding interpretation of the MERFISH kidney data. We are also grateful to Sandra Schmid, Rafael Gómez-Sjöberg, and Alejandro Granados for thoughtful comments on this manuscript. This work was supported by the Chan Zuckerberg Biohub. G Emanuel and J He own equity in Vizgen.

### Author Contributions

J Liu: conceptualization, resources, data curation, software, formal analysis, validation, investigation, visualization, methodology, and writing—original draft, review, and editing.
V Tran: conceptualization, resources, data curation, investigation, visualization, methodology, and writing—original draft.
VNP Vemuri: conceptualization, data curation, software, formal analysis, visualization, methodology, and writing—original draft, review, and editing.
A Byrne: conceptualization, resources, data curation, formal analysis, investigation, methodology, and writing—review and editing.
M Borja: conceptualization, resources, data curation, validation, investigation, visualization, methodology, and writing—review and editing.
YJ Kim: conceptualization, data curation, software, methodology, and writing—review and editing.

S Agarwal: conceptualization, data curation, software, formal analysis, validation, visualization, methodology, and writing—review and editing.

R Wang: conceptualization, resources, investigation, visualization, methodology, and writing—review and editing.

K Awayan: resources, data curation, software, formal analysis, validation, visualization, and writing—review and editing.

A Murti: resources, data curation, formal analysis, validation, methodology, and writing—review and editing.

A Taychameekiatchai: resources, data curation, validation, investigation, visualization, methodology, and writing—review and editing.

B Wang: resources, data curation, validation, investigation, visualization, methodology, and writing—review and editing.

G Emanuel: conceptualization, resources, data curation, software, formal analysis, validation, investigation, and methodology.

J He: resources, software, formal analysis, validation, visualization, methodology, and writing—review and editing.

J Haliburton: conceptualization, resources, data curation, software, supervision, investigation, methodology, project administration, and writing—review and editing.

A Oliveira Pisco: resources, data curation, software, formal analysis, supervision, validation, investigation, visualization, methodology, and writing—review and editing.

NF Neff: conceptualization, resources, data curation, supervision, investigation, visualization, methodology, and writing—review and editing.

## Conflict of Interest Statement

The authors declare that they have no conflict of interest.

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
