## [Reviewer comments · Life Science Alliance]

Life Science Alliance

Concordance of MERFISH Spatial Transcriptomics with Bulk and Single-cell RNA Sequencing

Jonathan Liu, Vanessa Tran, Venkata Naga Vemuri, Ashley Byrne, Michael Borja, Yang-Joon Kim, Snigdha Agarwal, Ruofan Wang, Kyle Awayan, Abhishek Murti, Aris Taychameekiatchai, Bruce Wang, George Emanuel, Jiang He, John Haliburton, Angela Oliveira Pisco, and Norma Neff

DOI: <https://doi.org/10.26508/lsa.202201701>

Corresponding author(s): Norma Neff, Chan Zuckerberg Biohub and Angela Oliveira Pisco, Chan Zuckerberg Biohub

Review Timeline:

Submission Date:	2022-08-31
Editorial Decision:	2022-09-01
Revision Received:	2022-09-13
Editorial Decision:	2022-09-14
Revision Received:	2022-09-27
Accepted:	2022-09-28

Transaction Report:

Please note that the manuscript was previously reviewed at another journal and the reports were taken into account in the decision-making process at Life Science Alliance.

Reviewer #1 Review

Report for Author:

The authors presented a systematic comparison of imaging-based MERFISH technology and the single-cell RNA-sequencing technology using samples from the kidney and the liver.

While the paper is appealing to the audience using MERFISH, I do not think the appeal is broad enough to reach other audiences such as users of ReadCor, SeqFISH, RNAscope, pciSeq. It would be good if the authors can include datasets from these other technologies in the comparison with scRNAseq.

The comparison with scRNAseq has been previously done in separate papers. In fact, comparing with scRNAseq is one of the first analyses that previous papers of MERFISH, SeqFISH have done to validate their own technologies. For example, in Moffit et al, Science, PMC6482113, Figure 4 C, D, E of Moffit paper have already performed comparison of scRNASeq and MERFISH, showing high degree of concordance at the bulk level. In Eng et al, Nature, the paper also performed comparison of SeqFISH+ with scRNASeq in Extended Figures 4, 5. The general agreement of MERFISH to scRNAseq is already known. Thus, it is unclear what the novelty of this paper is.

Another concern I have of this paper is the use of DAPI signal for segmenting cells often gives a very incomplete picture of mRNA distribution. DAPI is a nuclear stain. Transported RNAs are often found outside the nucleus. Thus, using DAPI as a boundary indicator will lead to a severe underestimate of total transcript count per cell. I would advice the authors try cell membrane stain such as actin.

The focus of the scRNAseq and MERFISH comparison is also limited to a set of genes profiled by MERFISH. This is both a strength and a weakness. The strength is that you almost always will get a high agreement with scRNAseq when genes profiled by MERFISH are differentially expressed cell type markers. The weakness with this kind of comparison is that the remaining 20,000-300 non-profiled genes are not compared. Perhaps the authors should design a better MERFISH experiment incorporating an unbiased list of genes, based on low, medium, high expression levels, rather than whether genes are differentially expressed for certain cell types. May be pick a few spatial genes as well. In my opinion, the comparison between MERFISH and scRNAseq should include a unbiased selection of genes.

Lastly, the authors integrated scRNAseq and MERFISH datasets. The choice of using Tabula Muris (TM) scRNASeq atlas is very puzzling. I would think MERFISH reflects a sample level picture, while TM reflects a population level picture. Naturally, you will expect the proportions of cell types to be different between the two. It is pre-mature to conclude that MERFISH more reliably identifies cell types alone than if it were integrated with scRNAseq. Authors should use serial section of kidney or liver slices, using one slice for MERFISH, and another slice for doing a separate scRNAseq experiment on it, for properly doing the integration.

Reviewer #2 Review

Report for Author:

Liu et al. provided a technical comparison between imaging-based MERFISH and sequencing-based single-cell technology. By measuring the 307 RNA profiles of mouse liver and kidney samples, this work systematically assessed the technical noise, reproducibility, single-cell RNA counts, measurement sensitivity, and MERFISH's ability of resolving distinct cell types. Their data indicated that computational integration with scRNA-seq reference datasets provides no substantial gain of signal, thus MERFISH data alone is sufficient to identify cell types and reproduce spatial structure. Overall, this work would be of interest in the field of spatial-omics technology and single cell analyses. However, the manuscript in current version contains several major issues and could be further improved by addressing the following questions:

Major comments:

1. The counterpart of MERFISH should be sequencing-based spatial technology such as Visium (10x Genomics), rather than scRNA-seq. Visium has been successfully applied on various tissue types including mouse liver and kidney. The significance of this work would be highly improved if the authors could provide systematic comparisons between the current MERFISH dataset and the published Visium spatial dataset.
2. Morphological H&E staining were totally missed. H&E-stained image from an adjacent tissue section was typically provided as a reference when analyzing spatial data. However, the reviewer could not find such information throughout the manuscript, especially for Figure 6 and Figure 7.
3. The MERFISH dataset was not fully utilized. There are 307 genes in the MERFISH panel, however, a majority of analyses only focused on the expression of cell-type marker genes. A comprehensive comparison of all the 307 genes between two modalities should be provided, categorized by gene ontology, such as cell type, cell cycle, metabolism, immune functions, etc.
4. In Figure 6 and Figure 7, the comparisons of cell type frequencies between the scRNA-seq reference, MERFISH data, and scANVI integration cannot represent the whole picture of the discrepancies. More detailed comparisons, such as cell states, functional features, signaling pathways in each identified cell type should be provided to draw any conclusion.

Minor comments:

1. In the first part of "Results" section, the detailed introduction of MERFISH's principle seems redundant since this is not the original contribution of the current work. Accordingly, I would suggest remove Figure 1 to supplementary figures.
2. For sequencing-based technology, the gene identification ability was highly relied on sequencing depth. Thus, the impact of sequencing depth on the comparison results should be evaluated and discussed.

Reviewer #3 Review

Report for Author:

The authors presented a technical study that compared RNA sequencing methods and MERFISH, on mouse liver and kidney tissues. The study provides a great opportunity to understand the strengths and weaknesses of each technique. It also highlighted the fact that MERFISH and scRNA-seq are very different methods with substantial and distinct technical bias that requires a careful integration strategy. While the paper is very well written, we have several questions that we think the authors should address before publication:

1. Can the authors address the observed discrepancy between MERFISH and scRNA-seq data? Why did MERFISH not detect pericytes, fibroblasts, or immune cells? Is it because the marker genes for these cell types are missing in the gene panel? Or is it due to these cells being discarded during QC (for example, could these cells have low DAPI staining?) What are the other possible reasons? Are there any cell types missing in scRNA-seq but detected in MERFISH (for example, due to dissociation induced artefacts in counting cell types)?
2. The authors performed correlation analysis of gene counts for MERFISH vs bulk RNA-seq (Fig 3c and 4d). Can the authors

also compute the correlation of gene counts in MERFISH vs scRNA-seq? For each gene (among only the cells where that gene is detected), what is the ratio of the median or mean count in the MERFISH data for that gene compared to the median or mean count in the scRNA-seq data? This should also give us a rough estimate for the detection efficiency for each gene.

3. In addition to cell type visualization (Fig 6 b,d and 7 b,d) and bar plots (Fig 6e and 7e), could the authors plot a heatmap showing the fraction of cells in each cell type for MERFISH vs cell types derived from integration with scRNA-seq data? This would show the extent of agreement in cell type annotation between MERFISH and scRNA-seq in a clearer fashion.
4. "trained a joint model using scVI's default". The default parameters are unlikely to be optimal. Could the authors try adjusting the scVI and scANVI parameters and check if the integration outcome can be improved?
5. Deep learning based integration is prone to overfitting. Can the authors try Seurat or Harmony integration? Or could the authors address whether there is any overfitting in their analyses?
6. To understand potential technical bias in Tabula Muris dataset, could the authors try integration with another or an additional scRNA-seq dataset?
7. Count statistics for MERFISH: scRNA-seq has complex technical noise, but the data can be modelled somewhat successfully with, for example, the zero-inflated negative binomial (ZINB) model. What distribution do the MERFISH gene counts follow? Could the authors propose an appropriate model for MERFISH data?
8. MERFISH bioinformatic analysis description is too brief and should not be lumped together with scRNA-seq: How were the MERFISH counts normalized? Were the counts per cell normalized by cell volume? Were there any other transformation (e.g., z-score) applied?
9. Can the authors described in more detail how was the cell segmentation performed? Were the transcripts localization used for cell segmentation or was only the DAPI image data used for the segmentation? Can the authors provide some estimate of the segmentation accuracy (for example, by comparing to some manually hand-drawn cells)?
10. Can the authors make the software or scripts used to perform data pre-processing and data integration available for download?

Minor comments

11. What are the primary and secondary antibody mix for staining the cell boundary? Could the authors provide the concentrations, durations, temperatures, and catalog numbers?
12. How did the authors perform the library amplification? What is the protocol used?
13. Could the authors report the number of cells in the scRNA-seq data used in the integration analyses?
14. Did the authors use expandable gel (i.e., expansion microscopy)? Why not, especially since overcrowding is an issue.
15. "transcripts assignable to cells was around 30-50%": this is throwing away a large amount of data. Can the authors consider trying alternative assignment and segmentation strategy, like Baysor (Petukhov, V., Xu, R.J., Soldatov, R.A. et al. Nat Biotechnol 40, 345-354 (2022))
16. Inconsistent definition of drop-out rate:
 - a. "this was defined as the fraction of genes with zero counts out of the whole 307-gene panel"
 - b. "defined as the fraction of genes in the panel possessing non-zero counts per single cell"
17. Were the overlapped spots between the optical Z sections removed? If so, how were they removed? This is important to prevent over-counting of the same gene that is present in different optical Z sections.
18. Could the authors provide the sequences of all the DNA probes that were used (encoding probes and readout probes)?
19. Consider softening the language regarding false positive controls: The use of pancreatic markers as false positive controls could be problematic because we don't know the ground truth regarding whether these genes are expressed at low level or not at all in kidney or liver tissue. (Perhaps the authors could check the bulk RNA expression of those pancreatic markers in kidney or liver tissue and report the values found for those genes in the bulk RNA seq data).

September 1, 2022

Re: Life Science Alliance manuscript #LSA-2022-01701-T

Norma Neff
Chan Zuckerberg Biohub

-
--

Dear Dr. Neff,

Thank you for submitting your manuscript entitled "Concordance of MERFISH Spatial Transcriptomics with Bulk and Single-cell RNA Sequencing" to Life Science Alliance. The manuscript was assessed by expert reviewers at another journal and then transferred to LSA. We are interested in the findings and would like to invite further consideration of this manuscript at LSA pending the following revisions:

- Address Reviewer 2 and 3's comments

Thank you for this interesting contribution to Life Science Alliance. We are looking forward to receiving your revised manuscript.

Sincerely,

B. MANUSCRIPT ORGANIZATION AND FORMATTING:

Author Response:

We thank the reviewers for the comments, and have written a point-by-point response below in red text. In addition to responding to the reviewer remarks, we have also updated the manuscript and data with major improvements. Most notably, the single-cell data now includes an improved cell segmentation that provides much more accurate single-cell RNA counts. This has resulted in significantly reducing the discrepancy between manually annotated MERFISH single cell results and automatically annotated results from scVI and scANVI integration. In addition, the MERFISH data now reflect lower dropout rates and higher sensitivity compared to scRNA-seq, as shown in the updated Figure 4. Finally, we have created a new Figure 5 in response to Reviewer 3's comments, which highlight the per-gene statistical similarity between MERFISH and scRNA-seq data among cells with nonzero counts. Changes in the main text and supplement have been indicated with red text.

In summary, our changes include:

- Updated single-cell segmentation
- Modifications to all main text figures with new data and/or analysis
- Figure 1 moved to the supplement
- Slight modifications to main text to reflect changes in narrative
- Additions to discussion to address points brought up by reviewers
- New Figure 5 exploring single cell statistical distributions of RNA counts
- New supplementary figure exploring integration with Harmony and Symphony
- New supplementary figure exploring systematic differences between RNA statistics in MERFISH and scRNA-seq for some cell types
- Modifications to Methods section to clarify methodology

Finally, we have updated the data shared on Figshare as well as the Github repository containing the code used to generate the figures. An interactive data portal to explore our results is available at <https://spatial-transcriptomics.ds.czbiohub.org/>.

Summary of major changes:

- Updated main text figures with new data and analysis
- Moved old figure 1 to the supplement

- Created new figure 5 on single cell distributions and statistics
- Created new SI figure and section on using harmony and symphony to integrate MERFISH and scRNA-seq data
- Removed old SI figures 3 and 6 as they were not quantitatively meaningful
- Removed one kidney dataset from single-cell analysis as there were not enough RNA counts to be reliable
- Updated segmentation
 - Improved segmentation accuracy
 - Detected some cell types that we couldn't before
 - Improved integration results with single cell transcriptomics led to better agreement between manual annotation and automated label transfer
- Adding scaling to bioinformatics methodology

Reviewer #2:

Liu et al. provided a technical comparison between imaging-based MERFISH and sequencing-based single-cell technology. By measuring the 307 RNA profiles of mouse liver and kidney samples, this work systematically assessed the technical noise, reproducibility, single-cell RNA counts, measurement sensitivity, and MERFISH's ability of resolving distinct cell types. Their data indicated that computational integration with scRNA-seq reference datasets provides no substantial gain of signal, thus MERFISH data alone is sufficient to identify cell types and reproduce spatial structure. Overall, this work would be of interest in the field of spatial-omics technology and single cell analyses. However, the manuscript in current version contains several major issues and could be further improved by addressing the following questions:

Major comments:

1. The counterpart of MERFISH should be sequencing-based spatial technology such as Visium (10x Genomics), rather than scRNA-seq. Visium has been successfully applied on various tissue types including mouse liver and kidney. The significance of this work would be highly improved if the authors could provide systematic comparisons between the current MERFISH dataset and the published Visium spatial dataset.

We acknowledge the reviewer's point that comparing MERFISH with another spatial technology like Visium would provide an additional relevant basis of comparison. Although our main goal is to assess the single-cell comparison of MERFISH with scRNA-seq, we agree with the reviewer and have included two panels in the old Figure 3 where we compare bulk MERFISH results with publicly available bulk Visium results from mouse liver and kidney (Dixon et. al. 2022, Guilliams et. al. 2022). Because Visium is not a single-cell technology, any more fine-grained comparison would require either deconvolution of the Visium data or pseudo-coarse-graining of the MERFISH data, which we think to be outside the scope of this paper.

2. Morphological H&E staining were totally missed. H&E-stained image from an adjacent tissue section was typically provided as a reference when analyzing spatial data. However, the reviewer could not find such information throughout the manuscript, especially for Figure 6 and Figure 7.

We respectfully point out that one of the main advantages of imaging-based spatial transcriptomic methods like MERFISH is that the acquired images can provide much of the same information as H&E stains and decrease the importance of capturing an H&E stain from an adjacent tissue section. Here, our single-cell segmentation is based on a combination of nuclear DAPI and cell membrane protein stains, which provide similar morphological information to the traditional H&E method. Because our protein stains are used in the same section as the MERFISH technique, we possess imaging-based morphological information from the same sample as that of the RNA itself. Although our analysis focused on downstream bioinformatic comparisons, we have released all of the raw images as well on AWS (<s3://czb-tabula-muris-senis/spatial-transcriptomics/MERFISH-data/>) and envision that interested users can investigate them at will.

That said, we do acknowledge that much of the image presentation in the original manuscript was lacking. We have made substantial improvements to the images in the main text figures that we think provide a much more comprehensive morphological representation of the data than before.

3. The MERFISH dataset was not fully utilized. There are 307 genes in the MERFISH panel, however, a majority of analyses only focused on the expression of cell-type marker genes. A comprehensive comparison of all the 307 genes between two modalities should be provided, categorized by gene ontology, such as cell type, cell cycle, metabolism, immune functions, etc.

Our experiment focused on the question of if a limited custom panel consisting only of cell-type marker genes was sufficient for cell-type identification. Thus, the genes in the panel were individually selected only to be marker genes without consideration for other aspects such as cell cycle, metabolism, immune functions, etc. Comparisons on the basis of these other factors therefore are at risk of overinterpretation, as the genes in our panel likely provide insufficient information on information not related to cell-type identification.

More generally, this reflects a fundamental limitation of FISH-based spatial transcriptomics, in that one can only draw conclusions based on the nature of the genes represented in the panel. Future works should be careful to include as many genes as relevant to the questions at hand as possible – for example, in this work a possibly more prudent choice of panel would have been able to replace some of the marker genes with genes relevant for other biological properties such as those mentioned by the reviewer.

To acknowledge these points, we have added these ideas to the discussion and have tightened up the language in the introduction to more explicitly focus on the question of cell-type identification.

4. In Figure 6 and Figure 7, the comparisons of cell type frequencies between the scRNA-seq reference, MERFISH data, and scANVI integration cannot represent the whole picture of the discrepancies. More detailed comparisons, such as cell states, functional features, signaling pathways in each identified cell type should be provided to draw any conclusion.

The previous versions of these figures used an old version of our segmentation pipeline, which was less reliable and produced larger discrepancies between the scRNA-seq reference, MERFISH, and scANVI integration. With the new single-cell segmentation, many of these discrepancies have been reduced. In addition for the liver data specifically, because the section was not of a whole liver, some of the discrepancies are likely due to missing cell types in the MERFISH data.

Nevertheless, because the gene panel only consists of cell-type markers, care should be taken not to overinterpret these results as the MERFISH data can only provide clear insights on conclusions specifically related to cell-type identification. Thus, the more detailed comparisons mentioned by the reviewer, although important, should not be conducted on this dataset out of consideration for the highly tailored nature of the gene panel.

We do acknowledge that previously we were too vague in our conclusions. Thus, we have tightened up the language in the manuscript to only cast these discrepancies in the light of cell type frequencies, without drawing any conclusions on additional biological aspects such as cell state or signaling pathways.

Minor comments:

1. In the first part of "Results" section, the detailed introduction of MERFISH's principle seems redundant since this is not the original contribution of the current work. Accordingly, I would suggest remove Figure 1 to supplementary figures.

This is a good point and we have moved Figure 1 to the supplement.

2. For sequencing-based technology, the gene identification ability was highly relied on sequencing depth. Thus, the impact of sequencing depth on the comparison results should be evaluated and discussed.

To clarify, since MERFISH is an imaging-based technology, sequencing depth here only plays a role in the quality of the reference scRNA-seq dataset, Tabula Muris Senis. Because the custom MERFISH gene panel was designed from the scRNA-seq reference, the ability of the MERFISH experiment to identify cell types is ultimately limited by the quality, and thus in some part sequencing depth, of the reference.

In our case, Tabula Muris Senis was sequenced at a very high depth (please refer to the supplementary material of the original publication where a detailed analysis of sequencing depth is presented) and we have confidence in the reliability of the gene panel. More generally, future MERFISH studies need to take the quality of the scRNA-seq reference into consideration, and we have added this point to the discussion.

Reviewer #3:

The authors presented a technical study that compared RNA sequencing methods and MERFISH, on mouse liver and kidney tissues. The study provides a great opportunity to understand the strengths and weaknesses of each technique. It also highlighted the fact that MERFISH and scRNA-seq are very different methods with substantial and distinct technical bias that requires a careful integration strategy. While the paper is very well written, we have several questions that we think the authors should address before publication:

1. Can the authors address the observed discrepancy between MERFISH and scRNA-seq data? Why did MERFISH not detect pericytes, fibroblasts, or immune cells? Is it because the marker genes for these cell types are missing in the gene panel? Or is it due to these cells being discarded during QC (for example, could these cells have low DAPI staining?) What are the other possible reasons? Are there any cell types missing in scRNA-seq but detected in MERFISH (for example, due to dissociation induced artefacts in counting cell types)?

We thank the reviewer for the insightful comments. Much of the discrepancy was due to our old cell segmentation, which has been dramatically improved in the new version of the manuscript. Pericytes and immune cells have now been detected in the mouse kidney. In the case of immune cells, we think the reduced frequency in the MERFISH data compared to the scRNA-seq reference is due to technical bias in the case of the latter to preferentially detect immune cells (Wu et al. 2019; Denisenko et al. 2020; Ding et al. 2020; Koenitzer et al. 2020; Slyper et al. 2020), and that the MERFISH data may be a more reliable measurement since it doesn't rely on dissociation techniques. Although fibroblasts still haven't been detected, they are also detected at extremely low abundance in the scRNA-seq reference (< 1%).

Nevertheless, the reviewer brings up important points for discussion. For example, our marker gene panel was designed using standard differential expression analysis based on cell type assignments – since then, many improved techniques have been designed to better design marker gene panels that could potentially improve our cell-type identification (Aevermann et al. 2021; Missarova et al. 2021; X. Chen, Chen, and Thomson 2022). Further investigation of QC is also important, as the ability to tie cell-type identification with morphological, imaging-based metrics such as cell shape or DAPI nuclear information could add an extra dimension of precision to single-cell analysis. Additionally, RNA counting with whole-cell segmentation in mature tissue can be tricky, due to ambiguity in determining cell boundaries, presence of extracellular RNA, and inefficient membrane protein stains. All of these potentially can play a

role in introducing noise into the downstream bioinformatic analysis and represent barriers in imaging-based spatial transcriptomics.

As for the reviewer's final point, MERFISH is not a great method for detecting novel cell types missing in scRNA-seq because the marker gene panel must be designed from the scRNA-seq reference data. Thus, MERFISH is great for detecting cell types expected from scRNA-seq, and would only detect new cell types by fortunate accident.

All of these points have been added to the discussion, which we believe have improved the manuscript.

2. The authors performed correlation analysis of gene counts for MERFISH vs bulk RNA-seq (Fig 3c and 4d). Can the authors also compute the correlation of gene counts in MERFISH vs scRNA-seq? For each gene (among only the cells where that gene is detect), what is the ratio of the median or mean count in the MERFISH data for that gene compared to the median or mean count in the scRNA-seq data? This should also give us a rough estimate for the detection efficiency for each gene.

This is a fantastic suggestion and we have added a new Figure 5 in the main text (between the old Figures 5 and 6). This new figure dives deeply into statistical analysis of single-cell RNA counts per-gene and includes a panel showing the correlation of mean gene counts in MERFISH vs scRNA-seq, as suggested by the reviewer. For the liver, there is a medium correlation with no apparent bias for either technology, whereas for the kidney the correlation is a bit worse with scRNA-seq providing systematically higher counts, which we believe is due to a combination of the marker gene panel being worse for the kidney due to overcrowding reasons, as well as segmentation in the kidney being more difficult.

3. In addition to cell type visualization (Fig 6 b,d and 7 b,d) and bar plots (Fig 6e and 7e), could the authors plot a heatmap showing the fraction of cells in each cell type for MERFISH vs cell types derived from integration with scRNA-seq data? This would show the extent of agreement in cell type annotation between MERFISH and scRNA-seq in a clearer fashion.

This is a great suggestion, and we have added a heatmap confusion matrix for liver and kidney between MERFISH and the scANVI integration as a supplementary figure, to more easily show the extent of agreement in cell type annotation.

4. "trained a joint model using scVI's default". The default parameters are unlikely to be optimal. Could the authors try adjusting the scVI and scANVI parameters and check if the integration outcome can be improved?

In addition to running scVI and scANVI, we have now also used Harmony and Symphony (see below response to Point 5). For both methods, most predicted cell type annotated agreed well with the manual annotations (Fig 6F and 7F). However, for some cell types, the integration

methods consistently underperformed, for example labeling kidney podocytes as endothelial cells.

Rather than attempting to vary method parameters to see if this could be improved, we reasoned that perhaps the integration methods were working as intended and the data itself were at fault. As a result, we have now created a new Supplementary Figure 7, which explores systematic differences between the MERFISH and scRNA-seq data that we believe prevent robust integration. For example, we discovered that kidney cells manually annotated as podocytes in the MERFISH data possessed RNA count profiles that were more similar to kidney endothelial cells in the scRNA-seq data than to podocytes in the scRNA-seq data. In contrast, liver periportal hepatocytes, which did integrate well, were much more similar to each other between MERFISH and scRNA-seq than to other cell types such as pericentral hepatocytes.

This indicates that integration of MERFISH and scRNA-seq data is a two-step process, dependent on both the integration method and the underlying similarity between MERFISH and scRNA-seq as technologies. In the case of our data, while most cell types possessed quantitatively similar RNA count profiles, some cell types such as podocytes were systematically different enough that integration would likely fail regardless of method.

5. Deep learning based integration is prone to overfitting. Can the authors try Seurat or Harmony integration? Or could the authors address whether there is any overfitting in their analyses?

The reviewer brings up a good point and we have created a new Supplementary Figure 6, which uses Harmony (Korsunsky et. al. 2019) and Symphony (Kang et. al. 2021) as an alternative integration and annotation prediction method to scVI and scANVI. We opted for Harmony based on available benchmarking results (Luecken et. al. 2022). In general the annotations predicted by Symphony were similar to those predicted by scANVI, quantified by the confusion matrix with the MERFISH manual annotations (Fig 6F and 7F; SI Fig 6). In addition, several cell types, such as podocytes and pericytes in the kidney, possessed the same discrepancies in both Symphony and scANVI. We explored this systematic error in more depth in the above answer to Point 4 and in the new Supplementary Figure 7.

6. To understand potential technical bias in Tabula Muris dataset, could the authors try integration with another or an additional scRNA-seq dataset?

We appreciate the reviewer concern. However, because the MERFISH custom gene panel was designed specifically using Tabula Muris Senis as a reference, we think that for technical analysis purposes, integration only makes sense using that specific dataset. scRNA-seq is known to have strong batch effects from dataset to dataset, and because our gene panel is tailor-made from differential expression analysis performed on Tabula Muris Senis, integration with a different scRNA-seq dataset is at risk of overinterpretation.

On a more general note, this highlights the highly specific nature of panel-based spatial transcriptomics techniques, and datasets like our MERFISH data here should only be interpreted in conjunction with their scRNA-seq reference, rather than as a general-purpose standalone dataset. We have added this point to the discussion for clarity.

7. Count statistics for MERFISH: scRNA-seq has complex technical noise, but the data can be modelled somewhat successfully with, for example, the zero-inflated negative binomial (ZINB) model. What distribution do the MERFISH gene counts follow? Could the authors propose an appropriate model for MERFISH data?

The reviewer brings up an excellent point and we have included this analysis in the new main text figure 5 mentioned above. In addition to comparing mean counts between MERFISH and scRNA-seq at the single-cell level, we also investigate the mean-variance relationship for each gene between MERFISH and scRNA-seq. Interestingly, we find that the two technologies possess remarkably similar statistics for these first two moments and conclude that the two follow similar distributions. For example, the negative binomial distribution successfully models genes for both scRNA-seq and MERFISH.

8. MERFISH bioinformatic analysis description is too brief and should not be lumped together with scRNA-seq: How were the MERFISH counts normalized? Were the counts per cell normalized by cell volume? Were there any other transformation (e.g., z-score) applied?

To address this point, we have elaborated the methods section to provide much more detail on the bioinformatic analysis and have separated the MERFISH bioinformatic analysis from the methodology for integration with scRNA-seq. To summarize, the technical comparisons in Figures 2, 4, and 5 looked at raw counts for both scRNA-seq and MERFISH. For the cell-type identification analysis in Figures 6 and 7, the MERFISH counts were normalized, log-transformed, and then scaled to unit variance and zero mean. We did not normalize by cell volume.

9. Can the authors described in more detail how was the cell segmentation performed? Were the transcripts localization used for cell segmentation or was only the DAPI image data used for the segmentation? Can the authors provide some estimate of the segmentation accuracy (for example, by comparing to some manually hand-drawn cells)?

The old cell segmentation made use of the StarDist segmentation algorithm on the DAPI image to identify the nuclei of each cell and then identified the maximum image intensity along each of several predetermined radial directions to identify the boundary of each cell. In the new version of the manuscript, we have switched to a much improved segmentation, which makes use of Cellpose (Stringer et. al. 2021). This method uses a combination of the DAPI nuclear and cell

boundary stain with the Cellpose cyto2 + nuclei algorithms to create a whole-cell segmentation. The detected transcripts were not used for segmentation. As can be seen in the old Figure 4 compared to the new Figure 3, the segmentation is much better (as seen by simple visual inspection).

10. Can the authors make the software or scripts used to perform data pre-processing and data integration available for download?

Yes – all of the Jupyter notebooks used to generate the figures will be available on the Github repository. The preprocessed and postprocessed data are available from Figshare, and the raw images are available on AWS. All links are included in the Data and Code availability section.

Minor comments

11. What are the primary and secondary antibody mix for staining the cell boundary? Could the authors provide the concentrations, durations, temperatures, and catalog numbers?

We utilized the Vizgen Cell boundary Staining Kit (Catalog number: 10400009), a protein-based staining reagent consisting of a Blocking Buffer Premix (PN: 20300012), a Primary Staining Mix (PN: 20300010, 0.5mg/ml) and a Secondary Staining Mix (PN: 20300011, 0.33mg/ml) that can label 3 plasma membrane proteins in cell. To perform the cell boundary staining, samples are first incubated with Blocking solution supplemented with Rnase inhibitor for 1 hour at room temperature. Then the sample is incubated with Primary Staining Solution diluted in Blocking Buffer (1:100 dilution) at room temperature for 1 hour. The sample is washed 3 times with 1X PBS, 5 minutes each time, and then incubated with Secondary Staining Solution diluted in Blocking Buffer (1:33 dilution) at room temperature for 1 hour. After washing with 1XPBS 3 times, 5 minutes each time, the sample is post fixed with 4% PFA at room temperature for 15 minutes. All of this information has been included in the Methods section.

12. How did the authors perform the library amplification? What is the protocol used?

Because MERFISH is a FISH-based technology, there was no library amplification performed, as that is only needed for sequencing-based assays. The set of probes used to imprint the binary barcodes onto the target transcripts was provided in ready to use format from Vizgen.

13. Could the authors report the number of cells in the scRNA-seq data used in the integration analyses?

Yes – we have included these numbers in the new version of the manuscript.

14. Did the authors use expandable gel (i.e., expansion microscopy)? Why not, especially since overcrowding is an issue.

We did not use expansion microscopy here. Although we discovered overcrowding to be an issue, it has not been a widely reported phenomenon in MERFISH studies, and so we did not conduct the experiments with this expectation. We hope that our detailed investigation of the overcrowding effect (see SI Figure 3) provides a useful reference for the field moving forwards.

For future experiments, we anticipate an increased need for expansion microscopy, especially as the number of genes in the panel increases. For this paper, we believe expansion microscopy to be outside the scope, especially as the technique has not been utilized in mouse liver or kidney in conjunction with MERFISH and would likely require much optimization.

15. "transcripts assignable to cells was around 30-50%": this is throwing away a large amount of data. Can the authors consider trying alternative assignment and segmentation strategy, like Baysor (Petukhov, V., Xu, R.J., Soldatov, R.A. et al. Nat Biotechnol 40, 345-354 (2022))

With the new segmentation, these numbers have dramatically improved, and ~70% of transcripts have now been assigned to cells. Because of the presence of extracellular RNA, we do not expect this number to ever reach 100%. We opted not to use a transcript-based segmentation strategy like Baysor because we wanted our segmentation information to be decoupled from the transcript information. Especially for a tissue like mouse kidney, which has been relatively understudied with MERFISH, we believe this decoupling risks less bias, as the image segmentation can always be improved, whereas an indirect method that relies on assumptions about transcript positions runs the risk of generating a self-consistent but perhaps biologically erroneous result.

16. Inconsistent definition of drop-out rate:

- a. "this was defined as the fraction of genes with zero counts out of the whole 307-gene panel"
- b. "defined as the fraction of genes in the panel possessing non-zero counts per single cell"

We thank the reviewer for pointing this out and have fixed this discrepancy in the main text.

17. Were the overlapped spots between the optical Z sections removed? If so, how were they removed? This is important to prevent over-counting of the same gene that is present in different optical Z sections.

The measurement was acquired with optical sectioning with z sections separated by 1.5um using a 1.4NA objective to minimize overcounting. Overlapping spots were removed by removing duplicate detections of a transcript when the same transcript was detected at the same X and Y position in adjacent z slices.

18. Could the authors provide the sequences of all the DNA probes that were used (encoding probes and readout probes)?

While we are unable to share the sequences of all the DNA probes used, we are able to share the specific subregions of each probe that bind to the target RNA transcripts, to allow readers to understand which specific regions are being targeted. We have included a list of all of these sequences as a supplementary file available for download on Figshare.

19. Consider softening the language regarding false positive controls: The use of pancreatic markers as false positive controls could be problematic because we don't know the ground truth regarding whether these genes are expressed at low level or not at all in kidney or liver tissue. (Perhaps the authors could check the bulk RNA expression of those pancreatic markers in kidney or liver tissue and report the values found for those genes in the bulk RNA seq data).

We think our analysis holds, since we have reported the detected RNA counts for these pancreatic markers for both the MERFISH and scRNA-seq data. If we believe in the validity of the scRNA-seq reference, then it should serve as a legitimate basis of comparison. This belief is validated because the expression levels of the pancreatic markers are incredibly low for both MERFISH and scRNA-seq (see the new Figure 4). Because we investigate only cells that have nonzero counts for either MERFISH or scRNA-seq, the scRNA-seq data should provide a higher quality basis than bulk RNA-seq data, which is unable to filter out cells with zero expression like we do here.

That said, we acknowledge that our language was perhaps too strong and have modified it in the text to now reflect this. We now refer to these as “effective false positive” controls.

September 14, 2022

RE: Life Science Alliance Manuscript #LSA-2022-01701-TR

Dr. Norma F. Neff
Chan Zuckerberg Biohub
Genomics Platform
499 Illinois Street
4th Floor
San Francisco, CA -94158

Dear Dr. Neff,

Thank you for submitting your revised manuscript entitled "Concordance of MERFISH Spatial Transcriptomics with Bulk and Single-cell RNA Sequencing". We would be happy to publish your paper in Life Science Alliance pending final revisions necessary to meet our formatting guidelines.

- please add ORCID ID for both corresponding authors-you should have received instruction on how to do so
- please make sure that all authors that are listed in the manuscript are also entered in our system and that the author order in the manuscript and in our system match
- please add the author contributions and a conflict of interest statement to the main manuscript text
- please use the [10 author names, et al.] format in your references (i.e. limit the author names to the first 10)
- under the Mouse tissue sample section of the Materials and Methods, please indicate approval for the animal work, and who granted that approval
- the Supplementary Text section should be incorporated into the main manuscript text

A. FINAL FILES:

B. MANUSCRIPT ORGANIZATION AND FORMATTING:

Sincerely,

September 28, 2022

RE: Life Science Alliance Manuscript #LSA-2022-01701-TRR

Dr. Norma F. Neff
Chan Zuckerberg Biohub
Genomics Platform
-499 Illinois Street
4th Floor
-San Francisco, CA -94158

Dear Dr. Neff,

Thank you for submitting your Resource entitled "Concordance of MERFISH Spatial Transcriptomics with Bulk and Single-cell RNA Sequencing". It is a pleasure to let you know that your manuscript is now accepted for publication in Life Science Alliance. Congratulations on this interesting work.

DISTRIBUTION OF MATERIALS:

Again, congratulations on a very nice paper. I hope you found the review process to be constructive and are pleased with how the manuscript was handled editorially. We look forward to future exciting submissions from your lab.

Sincerely,
